# Multiple independent losses of the biosynthetic pathway for two tropane alkaloids in the Solanaceae family

Jiao Yang [1,4], Ying Wu[1,4], Pan Zhang[1,4], Jianxiang Ma[1,4], Ying Jun Yao[2,4], Yan Lin Ma[1], Lei Zhang[3], Yongzhi Yang[1], Changmin Zhao [1], Jihua Wu[1], Xiangwen Fang[1] & Jianquan Liu [1,2] ✉

Hyoscyamine and scopolamine (HS), two valuable tropane alkaloids of significant medicinal importance, are found in multiple distantly related lineages within the Solanaceae family. Here we sequence the genomes of three representative species that produce HS from these lineages, and one species that does not produce HS. Our analysis reveals a shared biosynthetic pathway responsible for HS production in the three HS-producing species. We observe a high level of gene collinearity related to HS synthesis across the family in both types of species. By introducing gain-of-function and loss-of-function mutations at key sites, we confirm the reduced/lost or re-activated functions of critical genes involved in HS synthesis in both types of species, respectively. These findings indicate independent and repeated losses of the HS biosynthesis pathway since its origin in the ancestral lineage. Our results hold promise for potential future applications in the artificial engineering of HS biosynthesis in Solanaceae crops.

Secondary metabolites present in plants are due to robust evolutionary selection[1–3]. Remarkably, identical plant secondary metabolites can be identified in distantly related lineages[4], indicating that natural selection had acted multiple times to generate these metabolites via biotic and abiotic stressors, including defense against pathogenic microbes and herbivores, interactions with pollinators, and protection against drought, ultraviolet (UV) radiation, and frost[3]. An alternative explanation for this is that these metabolic pathways could have originated in the most recent common ancestor of all related lineages, and this trait was retained throughout distantly related lineages but potentially lost in some closely related lineages due to adverse selection pressures or genetic drift during diversification[5–8].

Tropane alkaloids are members of the diverse pyrroline class of plant alkaloids identified by the presence of an 8-methyl 8-azabicyclo [3.2.1] octane (tropine) ring within their chemical structure[9]. Over 300 tropane alkaloids have been identified across diverse plant families, including Solanaceae, Convolvulaceae, Rhizophoraceae, and Erythroxylaceae and others[9,10]. Tropane alkaloids can be grouped into several main classes according to their biosynthesis and structural characteristics[9]. For instance, cocaine is derived from the Erythroxylaceae family, hyoscyamine and scopolamine (HS) are produced in Solanaceae, and calystegines are isolated from Convolvulaceae[1]. Recent research has demonstrated that these tropane alkaloids present in Erythroxylaceae and Solanaceae have independently evolved but exhibit functional convergence, as indicated through enzyme characterization[10–13]. Within the Solanaceae family, the production of HS is observed in four distinct phylogenetic and geographical lineages. The tribe Datureae originates from South America, while the Mandragorinae and Hyoscyaminae (including *Atropa*, which was previously classified within the sister group Lyciinae) are found in the

[1]State Key Laboratory of Herbage Improvement and Grassland Agro-Ecosystem, College of Ecology, Lanzhou University, Lanzhou, China. [2]Key Laboratory of Bio-Resource and Eco-Environment of Ministry of Education, College of Life Sciences, Sichuan University, Chengdu, China. [3]Key Laboratory of Ecological Protection of Agro-Pastoral Ecotones in the Yellow River Basin, National Ethnic Affairs Commission of the People's Republic of China, College of Biological Science & Engineering, North Minzu University, Yinchuan 750021 Ningxia, China. [4]These authors contributed equally: Jiao Yang, Ying Wu, Pan Zhang, Jianxiang Ma, Ying Jun Yao. ✉e-mail: liujq@nwipb.ac.cn

Qinghai-Tibet Plateau and its neighboring regions and the Anthocercideae occurs in Australia[1,9,11,14]. These alkaloids possess diverse applications and are frequently employed in nerve agent poisoning, Parkinson's disease, and neuromuscular disorder treatment, serving as the foundation for the production of several effective drugs[15–17]. In addition, racemic HS is an effective, safe, and inexpensive medicine highlighted by the World Health Organization[17]. However, the low global supply of HS isolated from corresponding plants remains challenging[9]. Therefore, comprehending the genetic evolution of HS production and pinpointing specific genes and mutations responsible for this production may assist in the engineering of biosynthesis to generate these two bioactive and economically valuable products[18].

For over a century, various HS-producing species have been employed to uncover each step of the HS biosynthetic pathway[19–21]. Currently, the entire HS biosynthetic pathway has been completely characterized within *Atropa belladonna*[22] (Supplementary Fig. 1). It is implied that 12 enzymes play roles in the biosynthesis of HS, which begins with the polyamine putrescine (**1**), derived from two initial amino acid precursors, ornithine or arginine[9,23]. **1** is subsequently methylated through the action of putrescine methyltransferase (*PMT*) to generate *N*-methylputrescine (**2**), which is then oxidized by *N*-methylputrescine oxidase (*MPO*) to produce 4-methylaminobutanal (**3**)[18,24–26]. **3** then undergoes spontaneous cyclization to generate an *N*-methyl-pyrrolinium (**4**) cation, which is converted to tropinone (**5**) through a pathway mediated by the polyketide synthase (*PYKS*)[27] enzyme and cytochrome P450 (*CYP82M3*)[28]. Tropinone reductase I (*TRI*) then transforms **5** to tropine (**6**)[29], the initial starting material for HS biosynthesis. Aromatic amino acid aminotransferase (*ArAT4*) catalyzes the formation of phenylpyruvic acid (**8**), employing phenylalanine (**7**) as a substrate[30]. Phenylpyruvic acid reductase (*PPAR*) reduces **8** to phenyllactate (**9**)[31], which subsequently is converted by UDP glycosyltransferase (*UGT1*) to phenyllactylglucose (**10**), the acyl donor for the formation of littorine (**11**)[32]. Littorine (**11**) is generated when **6** is condensed along with **10** in a step catalyzed by littorine synthase (*LS*). Littorine (**11**) undergoes P450-mediated (*CYP80F1*) rearrangement to generate hyoscyamine aldehyde (**12**)[32,33]. Hyoscyamine dehydrogenase (*HDH*)[34] converts **12** to hyoscyamine (**13**), which is converted to scopolamine (**14**) via a two-step epoxidation reaction driven by the catalytic action of hyoscyamine 6-hydroxylase (*H6H*)[35,36]. Among the 12 sequential steps required for HS biosynthesis, the final five steps, beginning with the action of *TRI*, are exclusive to HS production. In contrast, the initial four enzymes and their respective products are involved in the synthesis of diverse metabolites, including cocaine and calystegine[1]. Notably, tropine (**6**), produced by TRI, is a pivotal intermediate metabolite within the HS pathway[37]. In addition, *LS*, another essential gene, is critical for initializing the dedicated biosynthesis of HS[9].

The above recent advances have led to numerous scientific questions. Specifically, it is unclear if the same biosynthetic genes are used by four HS-producing lineages to generate HS. If these lineages adhere to an identical genetic pathway for HS synthesis, independent origins of HS biosynthesis likely necessitate convergent functions within multiple key genes. Alternatively, these secondary metabolites potentially originated from an ancestral pathway that has been lost in closely related lineages. Thus, we sought to assess these evolutionary alternatives of HS biosynthesis, two medicinal tropane alkaloids with scattered distributions in phylogenetically distant plants of the Solanaceae family.

In this work, we sequence the genomes of three HS-producing species, including *Brugmansia arborea*, *Anisodus tanguticus*, and *Mandragora caulescens*, from three distantly related lineages of the Solanaceae family: Datureae, Hyoscyaminae, and Mandragorinae, respectively. Additionally, we produce a high-quality genome assembly for the non-HS-producing species *Lycium chinense* in the tribe Lycieae that is closely related to the Hyoscyameae. Furthermore, we obtain the high-quality genomes from 13 other non-HS-producing species that represent multiple non-HS-producing lineages for comparison. Our primary focus is to assess the evolutionary histories of HS biosynthesis in these distantly related lineages of the Solanaceae family. We assess the genomic structures, searched for synteny blocks representing conserved evolution of all HS-related genes at each synthesis step, and test the presence of critical loci for HS biosynthesis enzymes via gain-of-function and loss-of-function mutations across two types of species. The characterization of these genes and key mutations are instrumental for engineering HS pathways in widely grown Solanaceae crops, including tomatoes and potatoes, for the effective acquisition of these two metabolites in the future.

## Results

### Sequencing and assembling of four high-quality genomes

A total of 126 Gb, 166 Gb, 104 Gb, and 221 Gb of Illumina short reads were produced, used to estimate genome sizes corresponding to 1,198 Mb, 1,517 Mb, 756 Mb, and 1408 Mb for *Anisodus tanguticus, Brugmansia arborea, Mandragora caulescens*, and *Lycium chinense*, respectively (Table 1, Supplementary Fig. 2, Supplementary Tables 1 and 2). We then employed two long-read sequencing technologies to generate high-quality genomes corresponding to these species. For *A. tanguticus*, a total of ~164 Gb (~105.8×) of Oxford Nanopore Technologies (ONT) long reads possessing an N50 length of 25 kb were acquired and employed for the construction of primary contigs with NextDenovo software (Supplementary Table 1). Following polishing of the short reads using NextPolish (v 1.2.0)[38] and conducting de-redundancy filtering with purge_haplotigs[39], we obtained the eventual *A. tanguticus* genome assembly, having a total length of 1249 Mb and a contig N50 of 23.80 Mb (Table 1, Supplementary Table 3), which was marginally larger than the predicted genome size. For the remaining three species, PacBio high-fidelity (HiFi) long-read sequencing was conducted. A total of 20.1×, 33.0×, and 35.6× HiFi reads were acquired and employed to create final contig assemblies for *B. arborea* (1548 Mb with contig N50 of 7.70 Mb), *M. caulescens* (712 Mb with contig N50 of 25.26 Mb), and *L. chinense* (1538 Mb with contig N50 of 2.99 Mb), respectively (Table 1, Supplementary Table 3). According to the Hi-C data, we clustered and organized the contigs into chromosomes. For each genome, more than 97.47%, 97.42%, 94.67%, and 98.57% of the total assembled sequences were anchored into 24 *A. tanguticus* chromosomes, 13 *B. arborea* chromosomes, 24 *M. caulescens* chromosomes, and 12 *L. chinense* chromosomes, respectively (Fig. 1a, Supplementary Tables 4–7). The four assembled chromosome-level genomes exhibited high congruence, as the strongest signals from the Hi-C data were clustered on the anticipated diagonal (Supplementary Fig. 3).

The genome assembly quality was assessed via several approaches. Over 98.78% of the Illumina short reads were accurately mapped to the four genomes. The sequenced transcriptome data also exhibited a high mapping rate ranging from 81.10% to 97.37% (Supplementary Table 8). The assembled transcripts were mapped to the genomes, and over 70.25% of transcripts accounted for more than half the length of each chromosome of each species (Supplementary Data 1). BUSCO analysis was conducted, uncovering that over 98% of BUSCOs could be completely retrieved from the three HS-producing species, and a 95% complete BUSCO score was identified in the non-HS-producing *L. chinense* (Supplementary Table 9). These results established the high accuracy, continuity, and comprehensiveness of the four genomes described in this study.

### Genome annotation

Using a combination of de novo, homology, and transcriptome-based approaches, 46,606, 54,946, 32,347, and 29,193 protein-coding genes were predicted within the genomes of *A. tanguticus, L. chinense, B. arborea*, and *M. caulescens*, respectively (Table 1, Supplementary

## Table 1 | Features of genome assemblies and annotations

| Category | A. tanguticus | L. chinense | B. arborea | M. caulescens |
|---|---|---|---|---|
| Sequencing | | | | |
| Platform | Nanopore | PacBio | PacBio | PacBio |
| Genome-sequencing depth (X) | 105.77 | 35.57 (valid) | 20.09 (valid) | 33 (valid) |
| Assembly | | | | |
| Estimated genome size (Mb) | 1198 | 1408 | 1517 | 756 |
| Assembled genome size (Mb) | 1249 | 1538 | 1548 | 712 |
| N50 of scaffolds (bp) | 49,959,515 | 132,783,878 | 121,782,173 | 28,080,016 |
| No. of contigs | 205 | 1,406 | 1,398 | 808 |
| N50 of contigs (bp) | 23,808,256 | 2,994,494 | 7,701,694 | 25,262,060 |
| GC content of the genome (%) | 37.05 | 37.66 | 35.22 | 35.18 |
| Anchored to chromosome (%) | 97.47 | 98.53 | 97.40 | 94.67 |
| Complete BUSCOs (%) | 98.10 | 94.90 | 98.10 | 98.30 |
| Annotation | | | | |
| Percentage of repeat sequences (%) | 65.68 | 70.22 | 79.0 | 70.11 |
| LTR rate (%) | 43.07 | 44.48 | 58.37 | 36.62 |
| No. of predicted protein-coding genes | 46,606 | 54,946 | 32,347 | 29,193 |
| Average gene length (bp) | 4846.05 | 3775.38 | 3880.88 | 5447.13 |
| Average CDS length (bp) | 1134.68 | 1017.84 | 1128.02 | 1282.79 |
| Mean exon/intron length (bp) | 216.06/872.93 | 236.46/834.45 | 230.10/705.45 | 222.87/875.65 |
| Mean exon number per gene | 5.25 | 4.30 | 4.90 | 5.76 |

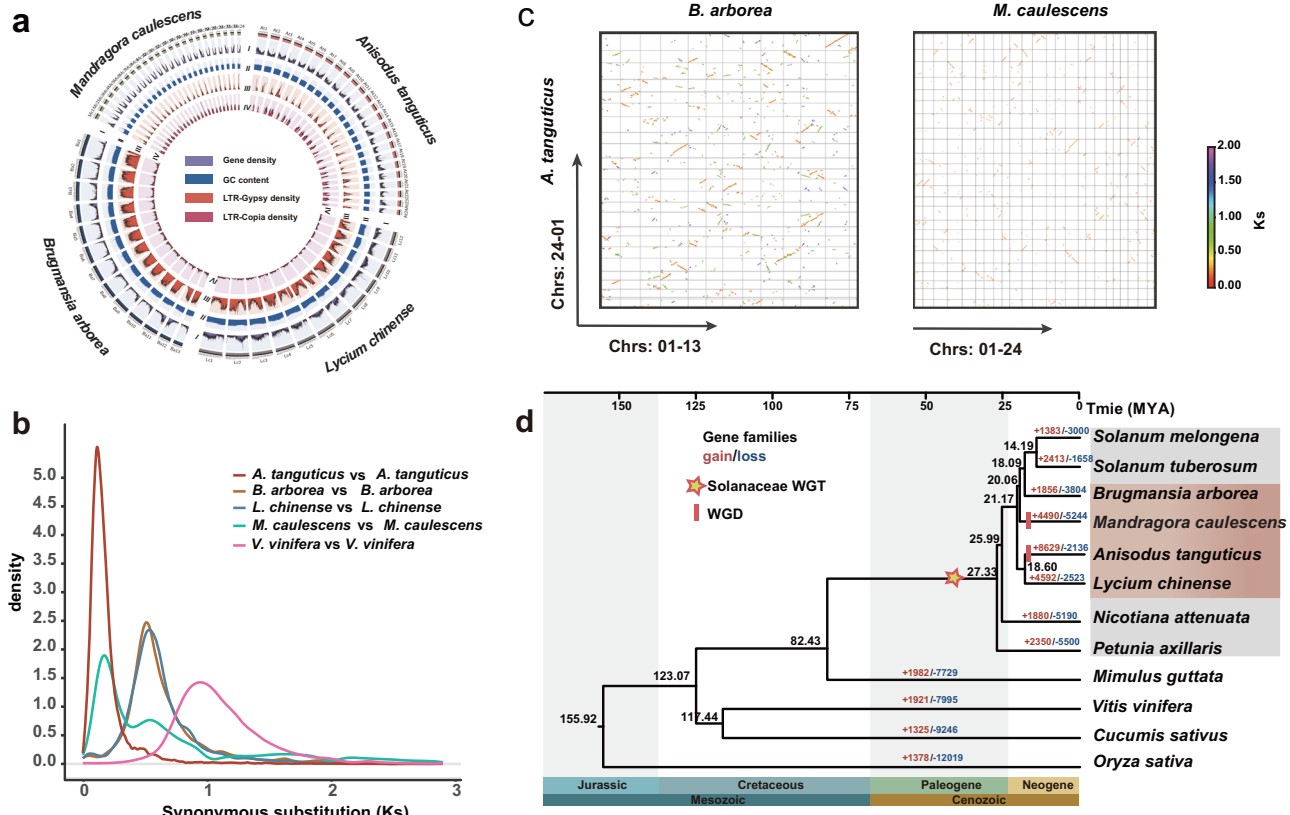

**Fig. 1 | A summary of the genomic characteristics of four Solanaceae species.** **a** The genome morphologies and Circos diagram for *M. caulescens*, *A. tanguticus*, *B. arborea*, and *L. chinense* are depicted. The different tracks (progressing inward) are as follows: (I) gene density; (II) guanine-cytosine (GC) content; (III) density of LTR-Gypsy transposons; and (IV) density of LTR-Copia transposons. **b** The distribution of synonymous substitution levels (*K*s) of syntenic paralogous genes. **c** A syntenic dot plot of comparison between *A. tanguticus* and *B. arborea*, and *M. caulescens*, respectively. **d** Phylogenetic tree of 12 species and evolution of gene families. The black numerical value alongside the node indicates the estimated divergence time of that particular node (MYA, million years ago). The number of gene-family expansion and contraction events (*p* value ≤ 0.01) is depicted in red and blue, respectively.

Tables 10–14, Supplementary Fig. 4). The completeness of the genome annotation employing BUSCO across the four gene sets was determined as follows: *A. tanguticus* (94.9%), *L. chinense* (92.2%), *B. arborea* (93.8%), and *M. caulescens* (93.1%) (Supplementary Table 15). For *A. tanguticus*, *L. chinense*, *B. arborea*, and *M. caulescens*, approximately 96.0%, 92.0%, 95.9%, and 98.56% of the identified protein-coding genes were successfully annotated using at least one database (including Swiss-Prot, KEGG, InterPro, Pfam, GO, NR, and COG) (Supplementary Table 16). Compared to the other three species, the percentage of predicted repetitive elements was substantially elevated within the genome of *B. arborea* (65.68%, 70.22%, and 70.11% compared to 79.0%; Supplementary Table 17). Transposable elements (TEs), known to be the predominant form of repeats found in angiosperm genomes[40], represented the most abundant subtype of repeat throughout all of four genomes (Supplementary Table 18). Among TEs, long terminal repeats (LTRs) were the most abundant, with *B. arborea* containing the highest LTR content (58.37%) and *M. caulescens* possessing the lowest (36.62%) (Supplementary Table 18). Additionally, we classified 1,752, 2,884, 2,325, and 2,727 proteins as transcription factors (TF) or transcriptional regulators (TR) within the genomes of *A. tanguticus*, *L. chinense*, *B. arborea*, and *M. caulescens*, respectively (Supplementary Table 19). Diverse noncoding RNA (ncRNA) genes were identified in all four species (Supplementary Table 20).

## Genome evolution
We examine the history of polyploidization across three phylogenetically unrelated HS-producing species within the Solanaceae family. For each genome, we identified the distribution of synonymous substitutions per synonymous site ($Ks$) utilizing syntenic paralogs. We observed a minor $Ks$ peak of approximately 0.5 in *A. tanguticus*, which was present across all other Solanaceae species, which corresponds to the shared whole-genome triplication event present in this family[41,42] (Fig. 1b). Moreover, we detected a substantial recent $Ks$ peak at approximately 0.15 and 0.20 in *A. tanguticus* and *M. caulescens*, respectively, suggesting the presence of recent polyploidization events that are potentially lineage-specific (Fig. 1b, Supplementary Fig. 5).

To elucidate the polyploidization history across all four species, we performed a comparative genomic assessment utilizing *Vitis vinifera* along with other Solanaceae species as placeholders. We detected a 6:1 syntenic depth ratio between *A. tanguticus* and *V. vinifera*, and a ratio of 2:1 between *A. tanguticus* and *B. arborea*, *L. chinense*, and *Solanum tuberosum*, respectively (Supplementary Figs. 6–8). Similarly, we characterized 2:1 syntenic depth ratios in our comparisons between *M. caulescens* - *L. chinense* and *M. caulescens*−*S. lycopersicum* (Supplementary Fig. 6), and a 2:1 depth ratio between *M. caulescens* and *S. melongena* and *C. annuum*, respectively (Supplementary Fig. 9). These findings confirmed that *A. tanguticus* and *M. caulescens* underwent a common ancestral whole-genome triplication (WGT) event alongside other Solanaceae species, but also experienced lineage-specific whole genome duplication (WGD) events. The syntenic dot plot between *A. tanguticus* and *M. caulescens* indicated that each fragment in each species could be identified with the two most related synteny fragments in another species, verifying that the recent WGD events were not shared between these two species (Figs. 1b, c).

A total of 12 species (*Oryza sativa*, *Cucumis sativus*, *V. vinifera*, *Mimulus guttata*, *Petunia axillaris*, *Nicotiana attenuata*, *L. chinense*, *A. tanguticus*, *M. caulescens*, *B. arborea*, *S. tuberosum*, and *Solanum melongena*) were chosen for gene family development. Nearly 404,865 genes were grouped into 26,783 gene families, and 220 single-copy gene families were characterized (Supplementary Figs. 10 and 11). According to these single-copy genes, a phylogenetic tree and divergence time were inferred (Fig. 1d, Supplementary Fig. 12). Our findings indicated that *P. axillaris* underwent the earliest divergence within the Solanaceae family, approximately 27.33 million years ago (Mya). *B.*

*arborea* was the *Solanum* species sister group, and both were closely related to *M. caulescens*, *A. tanguticus*, and *L. chinense*, encompassing a well-supported lineage that diverged from the *B. arborea* - *Solanum* lineage approximately 21.17 Mya (Fig. 1d).

We examined gene family expansion and contraction using CAFÉ. Expanded gene families and unique gene families were recognized within *A. tanguticus*, *L. chinense*, *B. arborea*, and *M. caulescens*, with 8,629/1085, 4,592/620, 1,856/2534, and 4,490/533, respectively (Fig. 1d, Supplementary Fig. 11). We identified that the expanded and unique genes were predominantly involved in heterocycle biosynthetic process, heme transporter activity, ADP binding, response to stimulus, and protein serine/threonine kinase activity through GO analysis of three HS-producing species (*A. tanguticus*, *B. arborea*, and *M. caulescens*), and in the establishment of localization substance metabolic process and nucleic acid binding through GO analysis of the non-HS-producing *L. chinense* (Supplementary Figs. 13–16).

## The three HS-synthesis species have a common HS biosynthetic pathway
Given that *A. tanguticus*, *B. arborea*, and *M. caulescens* all generate HS, it is possible that they employ a conserved pathway with a common evolutionary history to produce these compounds. To assess this further, we conducted a comparative genomic analysis, intending to classify the gene sequences responsible for HS synthesis in these species, through the use of a combination of BLAST search and HMM methods alongside the sequences functionally tested in *Atropa belladonna* as a reference, a species within the same lineage as *A. tanguticus*. We manually corrected erroneous automatic annotation and categorized all homologous genes into 12 enzyme families (Fig. 2a) with high sequence identity (>80%), low mismatches, and identical domains between the three species and *At. belladonna* (Supplementary Table 21, Supplementary Fig. 17). We also conducted a comparison between the identified genes and those isolated from three species that produce HS, functionally validated in previous studies[24,25,28,29,32,33,35,43]. We identified a high level of conservation with respect to both sequence similarity and gene expression patterns among these genes (Fig. 2a), indicating that these species should share a common HS biosynthetic pathway. To gain deeper insights into the function of these genes in HS biosynthesis within plants, we selected four key genes from the HS biosynthetic pathway, including *TRI*, *LS*, *HDH*, and *H6H*, for validation in *A. tanguticus* employing a preliminary virus-induced gene silencing (VIGS) technique established for this alpine plant. The findings of our experiments suggested that suppressing the expression of the *TRI*, *LS*, and *H6H* genes resulted in anticipated reductions in corresponding products, namely tropine (**6**), littorine (**11**), anisodamine (**15**), and scopolamine (**14**), in the roots of *A. tanguticus* (Supplementary Fig. 18). We, therefore, concluded that the identified candidate genes are required for HS biosynthesis in the species that produce HS. We detected possible gene clusters containing the 12 enzymes within each HS-producing species and determined that only *CYP82M3* and *TRI* clustered together in *A. tanguticus* and *B. arborea*, while the remaining genes were isolated from one another at different genomic loci (Fig. 2b). We further examined the expression levels of the identified genes across different tissues of the three HS-producing species (leaf, root, secondary root, and stem) and identified that all candidate genes were expressed in the roots and leaves of all three species (Fig. 2a). Notably, the expression of these genes was especially elevated in the secondary roots of *A. tanguticus*, contributing to the high concentrations of HS isolated from these roots, as observed in this group of species[1].

## Conserved syntenic blocks of HS-synthesis genes throughout the Solanaceae family
Evidence of loss of an ancestral genetic pathway can be offered by retained syntenic blocks corresponding to the key genes for a

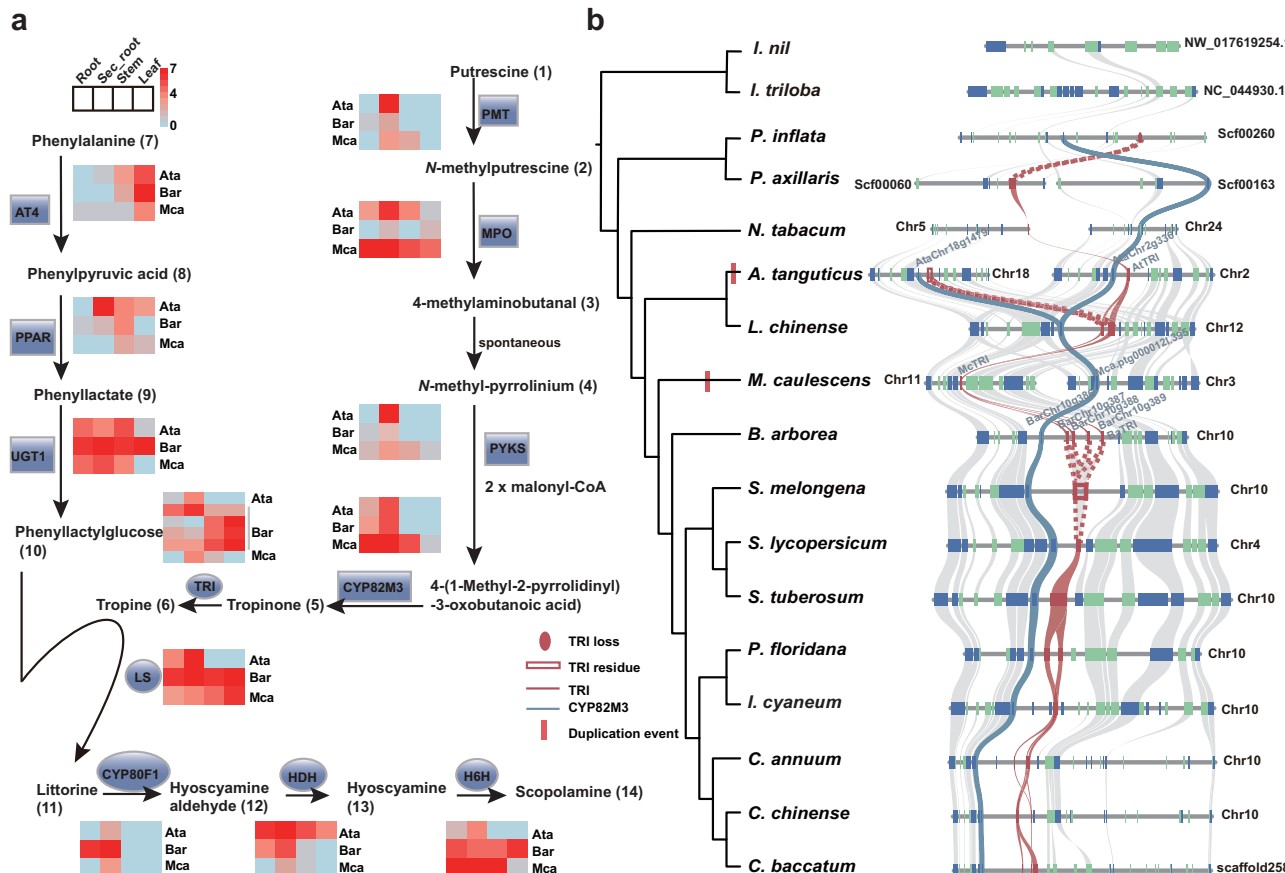

**Fig. 2 | HS biosynthesis, expression profiles, and microsynteny of two HS-synthesis genes. a** Overview of HS biosynthetic pathway and heat map of expression profiles for candidate genes from *A. tanguticus* (Ata), *B. arborea* (Bar), and *M. caulescens* (Mca), respectively. Four tandemly duplicated TRI genes from *B. arborea* with sequence similarity exceeding 90%. Abbreviation, sec_root: secondary root. **b** Phylogenetic tree for 15 selected species from the Solanaceae with *Ipomoea trilobal and Ipomoea nil* of Convolvulaceae used as external references (outgroup). The red bar indicates the WGD event in *A. tanguticus* and *M. caulescens* (left). The microsynteny analysis of TRI regions was identified across 17 species. Rectangles represent annotated genes, with orientation on the reverse strand (green) and same strand (blue). *TRI* gene names of *A. tanguticus*, *B. arborea*, and *M. caulescens* were marked above the gene blocks. The lines linking the syntenic *TRI* genes are highlighted in red and *CYP82M3* genes are highlighted in blue. The gray lines represent the gene collinearity across candidate species. The red dashed lines represent the assumed collinearity of incomplete *TRI* genes (pseudogene) from *A. tanguticus* and eggplant (right). Source data are provided as a Source Data file.

particular trait in closely related species lacking such a trait[5–8]. We identified highly conserved blocks in 11 out of 12 HS-synthesis genes, including *PMT*, *MPO*, *PYKS*, *CYP82M3*, *AT4*, *UGT1*, *TRI*, *LS*, *CYPB8OF1*, *HDH*, and *H6H* gene families throughout all three distantly related HS-generating species. Furthermore, 10 out of these 11 genes, with the exception of *HDH*, had syntenic genes and homologous complete or incomplete genes in at least one non-HS-producing species of Solanaceae (Figs. 2–4, Supplementary Figs. 19–27). For example, syntenic blocks of the downstream key gene *TRI* were recognized across all Solanaceae species, containing tandemly duplicated or incomplete orthologous *TRI* fragments in all species. Chromosome 2 of the HS-producing species, *A. tanguticus* containing *AtTRI*, exhibited a high level of intra-species synteny to chromosome 18 (Supplementary Fig. 23) with incomplete homologous *TRI*. The substitution rate ($K_s$) of the syntenic gene pairs between these two chromosomes was computed to determine the timescale of this duplication event, and a time of 13.41 Mya (95% CI: 7.93-18.89 Mya) was obtained (Supplementary Table 22). This time corresponded well to the species-specific WGD event in *A. tanguticus* (Fig. 1b). The *TRI* gene of this species originated prior to the specific WGD. The gene tree of *TRI* exhibited a similar arrangement to the species tree of the Solanaceae family (Supplementary Figs. 24, 28, 29), suggesting a relationship between the evolutionary history of *TRI* and the species of the family. Interestingly, when performing microsynteny analyses of *PPAR*, we could not

directly identify syntenic blocks between *B. arborea*, *A. tanguticus*, and *M. caulescens*, alongside non-HS-producing species, namely *Petunia inflata*, *P. axillaris*, and *L. chinense* (Supplementary Fig. 30a). However, via genome-wide gene family analyses, candidate orthologous genes were verified in these species (Supplementary Fig. 30b), indicating the potential of gene translocation and synteny erosion throughout lineage diversification. These factors might have underscored the observed differences in synteny[5]. The highly conserved syntenic blocks corresponding to the HS-synthesis genes, along with their phylogenetic relationships, indicate that the HS-synthesis pathways likely arose in the common ancestor of all Solanaceae lineages.

Compared to the other HS-synthesis genes exhibiting collinear conservation, we determined that three downstream HS-specific genes, *LS*, *CYP8OF1*, and *HDH*, displayed incomplete genic structure, resembling pseudogenes within non-HS-producing species (Fig. 4). For example, in one non-HS-producing species, *Iochroma cyaneum*, the first two exons of the *LS* gene underwent evolutionary changes, generating a pseudogene (Fig. 3a, b). Similarly, collinear pseudogenes were identified in non-HS-producing *Capsicum* species for the *CYP80F1* gene (Fig. 3d). While we could not identify the *HDH* residue in non-HS-producing species of Solanaceae, we confirmed its synteny blocks and complete genic structure in Convolvulaceae family species, belonging to the same Solanales order, sharing a close relationship (Supplementary Fig. 31). This indicates that the HDH gene possesses an

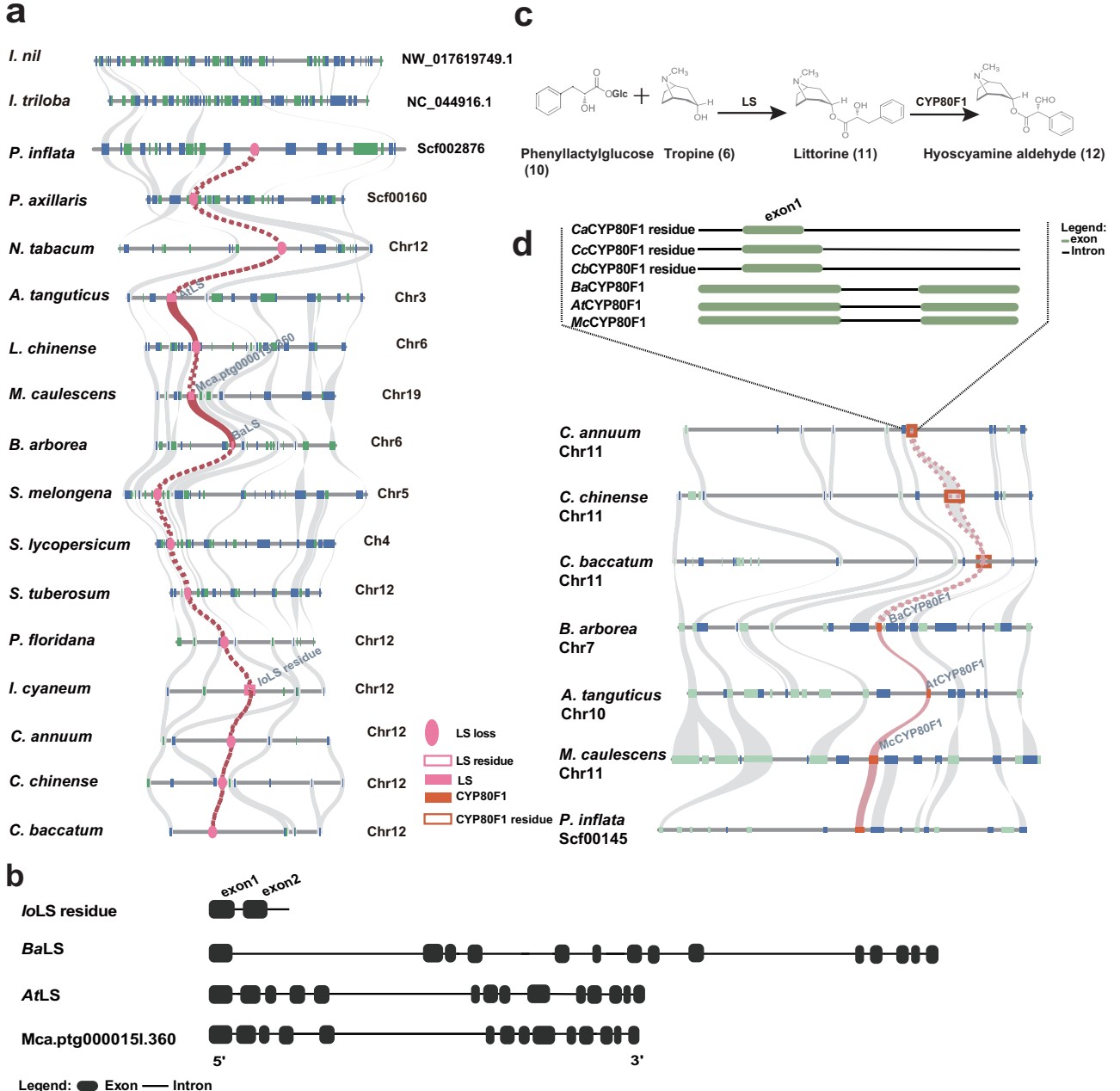

**Fig. 3 | Microsyntenty analysis of *LS* and *CYP80F1* regions. a** Microsynteny analysis of *LS* genes across 17 selected species. Rectangles represent annotated genes with orientation on the reverse strand (green) and same strand (blue). *LS* gene names from *A. tanguticus, B. arborea*, and *M. caulescens* were marked above the gene blocks. The lines associating the syntenic *LS* genes are highlighted in red. The gray lines indicate the gene collinearity across candidate species. **b** The position of LS residues of *Iochroma cyaneum* with black rounded rectangles indicating exons. **c** Biosynthetic pathway from the littorine and hyoscyamine aldehyde branch. **d** Syntenic assessment of *CYP80F1* genes between *P. inflata*, Pepper, *B.* *arborea, A. tanguticus*, and *M. caulescens. CYP80F1* gene names from *A. tanguticus, B. arborea*, and *M. caulescens* were indicated above the gene blocks. The black dotted rectangle indicates the position of CYP80F1 residues, while the green rounded rectangles in the dashed box indicate exons. The lines indicate genes having collinearity, and pink lines indicate colinear relationships of *CYP80F1* genes between *P. inflata, B. arborea, A. tanguticus*, and *M. caulescens*. The pink dashed lines signify the presumed collinearity of incomplete *CYP80F1* genes (pseudogene) across three *Capsicum* species.

ancestral origin predating the divergence between the two families. The absence of complete genic structures in non-HS-producing species for these genes may be linked to pseudogenization, resulting in a loss of exons (Fig. 3). Alternatively, these genes may exist in unsampled non-HS-producing species, which might be investigated in the future with the widespread availability of high-quality genomes. These results highlight the significant collinearity of HS-synthesis genes across three distantly related HS-producing species, alongside their partial or complete retention in non-HS-producing species as well as closely related families. Furthermore, the homologous genes situated at each locus from the three HS-producing species were determined to be paraphyletic, as opposed to monophyletic, as anticipated in cases of convergent evolution[44,45]. Therefore, the pseudogenization and loss of downstream HS-synthesis genes in collinear regions of non-HS-producing species have resulted in numerous losses of HS biosynthesis following a common recent origin (Fig. 4).

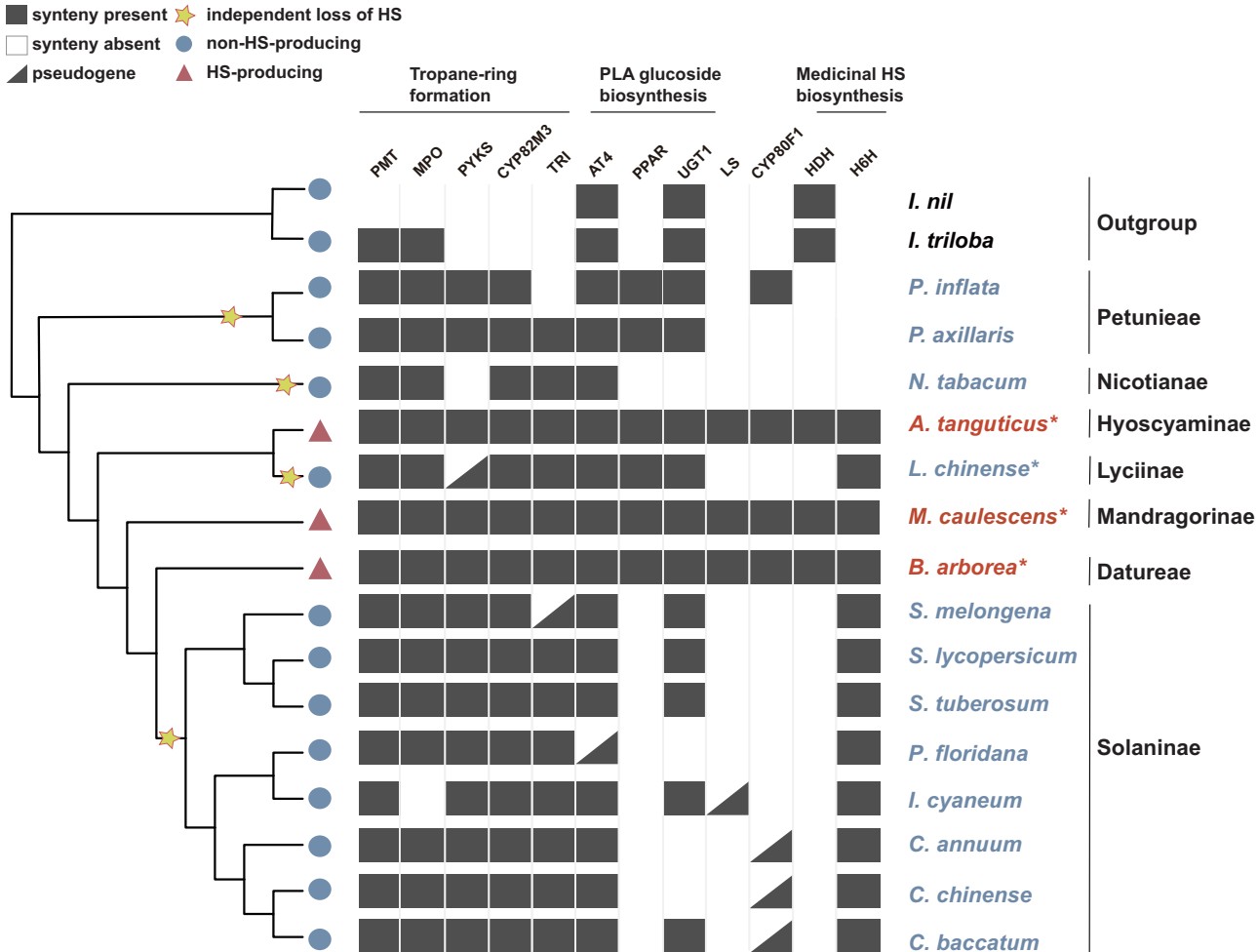

**Fig. 4 | Phylogenetic patterns of HS-associated genes in the Solanaceae family.** Phylogeny containing HS-producing species (red) and non-HS-producing (blue) species of Solanaceae family with *I. nil* and *I. triloba* used as outgroups. The absence, presence, or fragments (pseudogenes) of 12 HS genes are designated by white, black, and black triangle boxes, respectively. Stars indicate independent loss of the HS biosynthetic pathway. Asterisks indicate species that were sequenced for this study. Tropane-ring formation, the upstream genes of the HS-biosynthesis pathway; PLA glucoside biosynthesis, the branch genes of the HS biosynthesis pathway; Medicinal HS biosynthesis, the downstream genes of the HS biosynthesis pathway.

## Experimental tests of functional activities of the key gene *TRI*

The *TRI* gene is essential for the production of tropine (**6**), a critical intermediate metabolite in the HS synthesis pathway[9,37]. Using TRI as a representative gene, we intended to examine the hypothesis that HS-synthesis genes may have undergone functional reduction or loss throughout non-HS-producing species beginning from their ancestral origins. Compared to the other HS-synthesis genes, TRI has exhibited diverse patterns in non-HS-producing species, ranging from preservation in an intact form to evolution into a pseudogene or complete absence (Fig. 4). This unique variation makes TRI an optimal candidate for loss-of-function and gain-of-function experiments across multiple sites, incorporating mutual mutation experiments in both HS-producing and non-HS-producing species. By modulating *TRI*, we can gain insights into the functional magnitude of these genetic alterations and explore the role of HS-synthesis genes within the biosynthetic pathway. This approach is promising for elucidating the mechanisms underlying the loss or reduction of HS biosynthesis in non-HS-producing species over the course of evolutionary history.

We examined *TRI* sequences from 17 Solanaceae species (Supplementary Fig. 32) and identified high sequence identities, with 90% similarity among the three HS-producing species and 82% similarity across all species. We identified five variable sites (109, 155, 167, 201, and 243) critical for **5** limitation in the homologous TRI genes (Fig. 5a).

Although these sites were conserved across the HS-producing species, they exhibited variations in the non-HS-producing species due to relaxed evolution. We modeled the tertiary structure of the *B. arborea* *Ba*TRI protein (Fig. 5b) and identified a ternary complex containing *Ba*TRI, NADPH, and **5** via molecular docking. We found that Val[109] and Val[167] formed a hydrophobic pocket capable of binding tropinone. The distance between Val[109] and **5** was estimated to be 3.6–3.9 Å, while the distance between Val[167] and **5** was 3.8 Å. These two residues play stabilizing roles in positioning **5** in a preferred catalytic conformation via hydrophobic forces. Additionally, ancestral sequence reconstruction demonstrated a more than 99% probability for Val in both positions 109 and 167 (Supplementary Table 23).

We investigated NADPH-dependent **5** reduction reactions catalyzed by TRI proteins using the three HS-producing species (*A. tanguticus*, *B. arborea*, and *M. caulescens*) and the two non-HS-producing species (*L. chinense* and *S. tuberosum*) lacking the identified sites. The reduction reactions catalyzed by TRI were verified through ultra-performance liquid chromatography-mass spectrometry (UPLC-MS) (Supplementary Table 24). Comparison of MS spectra corresponding to the parent mass of **6** (m/z = 142.12) uncovered a peak at the retention time of 7.12 min in samples of *Ba*TRI (*B. arborea*), *At*TRI (*A. tanguticus*), *Mc*TRI (*M. caulescens*), and *Lc*TRI (*L. chinense*), which produced fragments with equal retention times to those identified

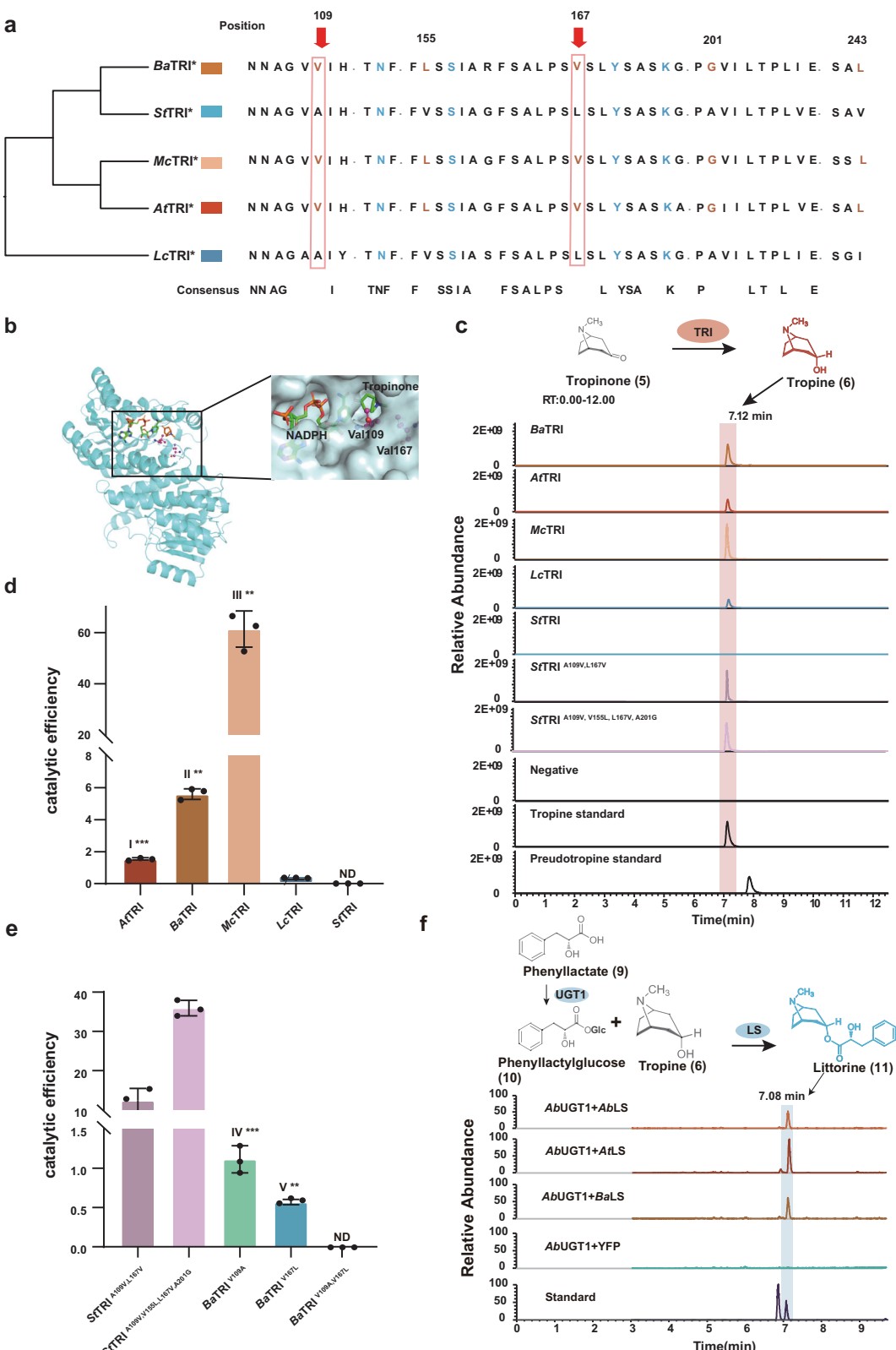

using a tropine (**6**) standard (Fig. 5c). The tropine (**6**) peak was undetected in *St*TRI (*S. tuberosum*) (Fig. 5c). The TRI activities of various species matched their recorded tropine (**6**) concentrations in each original species[9].

We generated Michaelis-Menten curves for the TRI enzyme alongside tropinone (**5**) substrates to determine $K$m and $V$max values, and used these values to deduce $K$cat and catalytic efficiency ($K$cat/

$K$m) values (Supplementary Fig. 33). Our examination revealed that *Lc*TRI had a 10-fold lower tropinone (**5**) affinity ($K$m = 2.91 ± 0.14 mM) than *Ba*TRI and *At*TRI ($K$m = 0.31 ± 0.03 mM and $K$m = 0.21 ± 0.01 mM, respectively) (Supplementary Table 25). However, an intriguing result emerged when investigating *Mc*TRI derived from the HS-producing species *M. caulescens*. *Mc*TRI displayed a significantly elevated affinity for tropinone, with a $K$m value of 0.09 ± 0.01 mM, demonstrating

**Fig. 5 | Characterized enzymatic activities of TRIs, TRI-mutants, and LSs from both HS-producing and non-HS-producing species. a** Comparison of critical amino acids of TRI implicated in HS-producing and non-HS-producing plant species (five TRIs from *M. caulescens* (*Mc*TRI), *B. arborea* (*Ba*TRI), *A. tanguticus* (*At*TRI), *L. chinense* (*Lc*TRI) and *S. tuberosum* (*St*TRI). Asterisks indicate enzymes confirmed through this study). The sequences highlighted in blue specify a catalytic tetrad. The orange highlights present crucial sites. Notably, sites 109 and 167 are also binding sites for tropinone. The location of the sites relative to the position in *Ba*TRI protein. Dots represent omitted sites in the sequences. **b** The tertiary structure of *Ba*TRI. Two purple ball-and-stick model bands represent essential sites for functional verification. **c** Extracted-ion chromatograms (EIC) indicating the in vitro activity of seven purified recombinant TRIs from *Ba*TRI, *At*TRI, *Mc*TRI, *Lc*TRI, *St*TRI, *St*TRI$^{A109V, L167V}$, and *St*TRI$^{A109V, V155L, L167V, A201G}$ with tropinone employed as the substrate. The evaluation of reduction catalytic efficiencies (*K*cat/*K*m values) of five wild-type (**d**) and five mutants TRIs (**e**). The raw Michaelis-Menten curves of these TRIs for the NADPH-dependent reduction reaction of tropinone are presented in Supplementary Fig. 33. Error bars are presented as mean ± SD of *n* = 3 biologically independent experiments. The different numerals in (**d**) and (**e**) indicate significantly different values to each other at $P < 0.001$ (\*\*\*) analyzed using a two-tailed Student's *t* test. ND not detectable. I\*\*\* ($p = 0.0009$), II\*\* ($p = 0.0013$), III\*\* ($p = 0.0044$), IV\*\*\* ($p = 0.0002$) and V\*\* ($p = 0.0013$) represent the significantly different values compare to *At*TRI and *Lc*TRI; *Ba*TRI and *Lc*TRI; *Mc*TRI and *Lc*TRI; *Ba*TRI$^{V109A}$ and *Ba*TRI; *Ba*TRI$^{V167L}$ and *Ba*TRI, respectively. **f** Extracted-ion chromatograms showing the in vivo activity of three *LS* genes from *A. tanguticus* (*At*LS), *B. arborea* (*Ba*LS), and *A. belladonna* (*Ab*LS) through co-expressing of the *AbUGT1* gene, responsible for the conversion of phenyllactate into phenyllactylglucose in tobacco leaves. Phenyllactate and tropine were used as substrates via co-infiltration into tobacco leaves, and the extracted ion chromatograms of hyoscyamine and littorine are illustrated in light blue. Source data are provided as a Source Data file.

exceptional catalytic efficiency (*K*cat/*K*m) values of 61.46 ± 5.76 (Supplementary Table 25, Supplementary Fig. 33). These values represent the maximum affinity reported thus far in any orthologous gene[1,9], suggesting that *McTRI* is a promising candidate for future metabolic engineering of HS alongside other tropane alkaloids. Furthermore, the catalytic efficiencies (*K*cat/*K*m values) of *At*TRI, *Ba*TRI, and *Mc*TRI with tropinone (**5**) (1.55, 5.60, and 61.46, respectively) were significantly more rapid and robust ($p = 0.0009$, $p = 0.0013$ and $p = 0.0044$, $p < 0.01$, *t*-test) compared to *Lc*TRI (0.24). Conversely, *St*TRI from potatoes exhibited no activity (Fig. 5d, Supplementary Table 25).

To test our hypothesis that specific sites in the *TRI* gene of non-HS-producing species may exhibit reduced or abolished activities due to relaxed evolution, we performed gain-of-function and loss-of-function mutation experiments on *St*TRI and *Ba*TRI, respectively, derived from non-HS-producing potato and HS-producing *B. arborea*. Through sequence analysis and a comprehensive literature review, it was determined that Val$^{109}$, Leu$^{155}$, Val$^{167}$, and Gly$^{201}$ residues in the *Ba*TRI protein were crucial for the maintenance of high activity. Subsequently, we performed key amino acid substitutions in *St*TRI and *Ba*TRI and examined their enzyme kinetics, including *St*TRI$^{A109V, L167V}$ and *St*TRI$^{A109V, V155L, L167V, A201G}$ in gain-of-function mutagenesis and *Ba*TRI$^{V109A}$, *Ba*TRI$^{V167L}$, *Ba*TRI$^{V109A, V167L}$ in loss-of-function mutagenesis. The mutants *St*TRI$^{A109V, L167V}$ and *St*TRI$^{A109V, V155L, L167V, A201G}$ displayed a significant increase in *K*cat/*K*m value, suggesting an enhanced catalytic ability for the conversion of tropinone (**5**) to tropine (**6**) (Fig. 5c, e; Supplementary Table 26 and Figure 33). In contrast, the *K*cat/*K*m values of *Ba*TRI$^{V109A}$ and *Ba*TRI$^{V167L}$ decreased significantly relative to *Ba*TRI ($p = 0.0002$ and $p = 0.0013$, $p < 0.01$, *t*-test). Interestingly, *Ba*TRI$^{V109A, V167L}$ had the ability to convert tropinone (**5**) to tropine (**6**) completely abolished (Fig. 5e, Supplementary Table 27 and Figure 33). These findings provide evidence that the key *TRI* gene in distantly related HS-producing species has maintained its ancestral active functions. In contrast, non-HS-producing species, while possessing the entire *TRI* gene (as observed in potatoes), have undergone reduced or even entire loss of functionality, probably due to relaxed evolution.

### Functional losses of critical downstream HS-synthesis genes

The functional losses of critical downstream HS-synthesis genes may have taken place via conserved evolutionary processes related to functional termination, gene pseudogenization, and gene loss in non-HS-producing species (Fig. 4), while these genes retain their ancestral functions in HS-producing species. For instance, we confirmed the activity of *LS* genes in catalyzing the conversion of tropine (**6**) and phenyllactylglucose (**10**) to littorine (**11**) in HS-producing species (Fig. 5f). However, we observed that specific non-HS-producing species still have collinear genes within the HS synthesis pathway downstream, namely *PPAR*, *CYP80F1*, *HDH*, and *H6H*. Therefore, investigating whether these genes from non-HS-producing species retain enzymatic functions is worthwhile.

To investigate the enzymatic function of PPAR, we described the functions of PPARs from *B. arborea*, *M. caulescens*, and *P. inflata* via in vitro enzymatic assays. Consistent with prior studies[22,46], all examined PPARs transformed phenylpyruvic acid (**8**) into phenyllactate (**9**) (Fig. 6a). With respect to CYP80F1 homologs, we cloned these genes and transiently expressed them in tobacco leaves. The results demonstrated that the *CYP80F1* gene of *P. inflata* (*PiCYP80F1*) lacked a complete functional domain in contrast with other HS-producing species (Supplementary Fig. 34). Then, we performed relative enzyme activity examination of four CYP80F1 homologs in *A. tanguticus* (*At*CYP80F1), *B. arborea* (*Ba*CYP80F1), *M. caulescens* (*Mc*CYP80F1), and *A. belladonna* (*Ab*CYP80F1) by co-expressing them transiently in tobacco leaves alongside *Ab*HDH, *At*HDH, *Ba*HDH, *Mc*HDH, *It*HDH (*I. triloba*), and *In*HDH (*I. nil*), which were cloned from their corresponding species individually to convert littorine (**11**) into hyoscyamine (**13**). After infiltrating with littorine (**11**), we measured littorine (**11**) and hyoscyamine (**13**) in all tobacco leaves expressing the corresponding construct (Fig. 6b), demonstrating the activity of all examined enzymes. Bioinformatics assessment and in vitro enzyme activity assays of the *H6H* gene indicated the enzymic activity of the *M. caulescens* (*McH6H*) and *L. chinense* (*LcH6H*) *H6H* genes (Fig. 6c, Supplementary Fig. 35). Prior studies have reported that the *H6H* gene of *A. tanguticus* and *B. arborea* catalyzed the two-step reaction for the production of scopolamine (**14**)[47,48]. Therefore, all of these results indicate that HS-synthesis pathways likely arose in the common ancestor of all Solanaceae lineages, and the loss or pseudogenization of these genes across some non-HS-producing species resulted in the limited distribution of HS across the Solanaceae family.

## Discussion

We successfully compiled four high-quality genomes for Solanaceae species, three of which produce HS, including *A. tanguticus*, *B. arborea*, and *M. caulescens*, and one which does not, *L. chinense*. Alongside previously reported genomes derived from non-HS-producing species, we discovered the evolutionary history of each genome and uncovered that both *A. tanguticus* and *M. caulescens* underwent an additional lineage-specific WGD event in comparison to other species, alongside the shared triplication event experienced by the entire Solanaceae family[49–51]. Although *L. chinense* did not undergo a lineage-specific WGD event, we observed the presence of high gene duplications within its genome. These duplications may result from the excessive retention of genes in this species from the ancestral WGT event for the total Solanaceae family or the other technical factors that needs further investigation in the future.

Our results indicate that three distantly related species within the Solanaceae family likely adhere to an identical HS biosynthetic pathway. In addition, we uncovered the role of *TRI*, *LS*, *PPAR*, *CYP80F1*, *HDH*, and *H6H* from HS-producing alpine species using VIGS and biochemical approaches. The in vitro method outlined above is effective

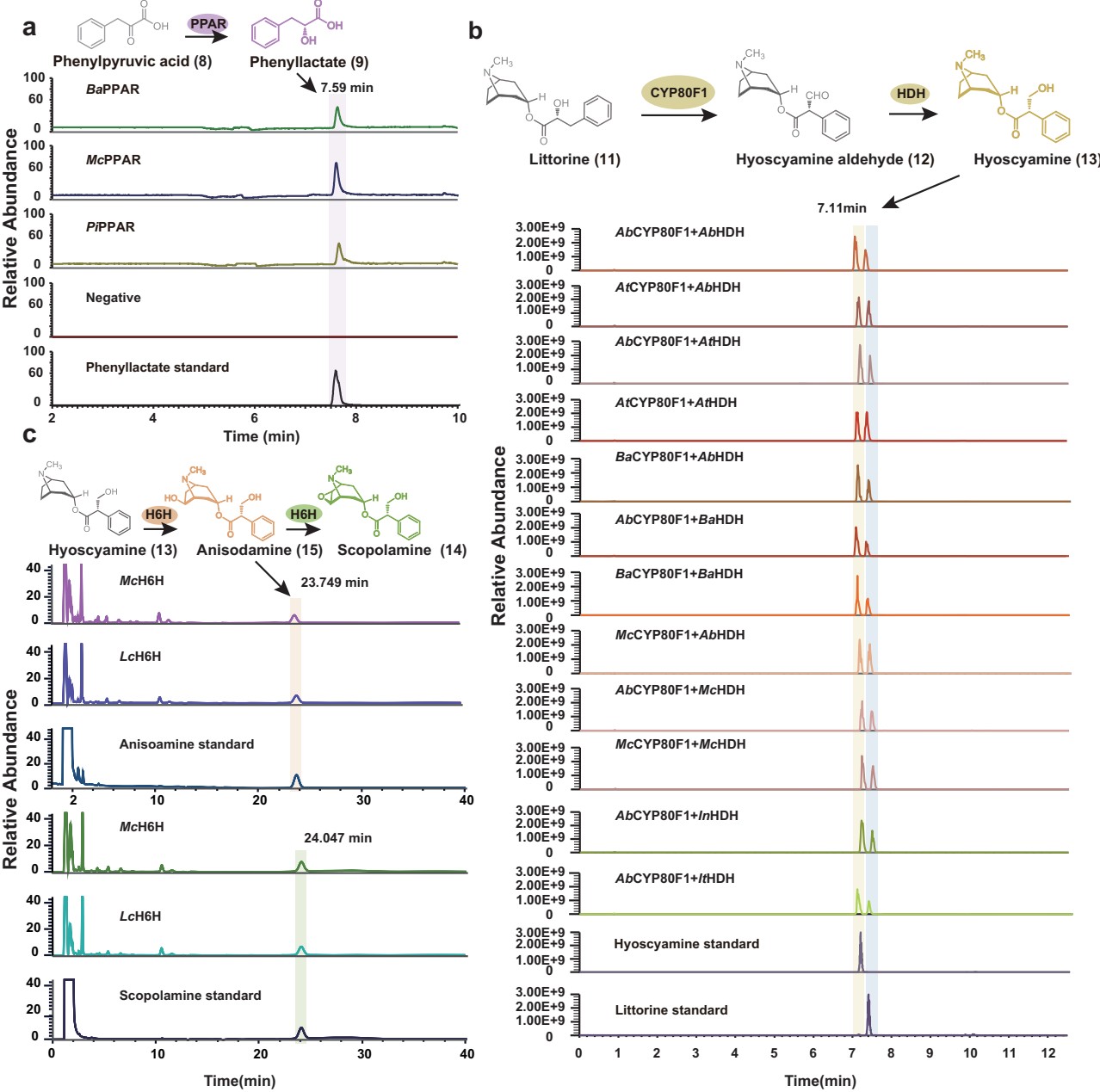

**Fig. 6 | Characterized enzymatic activities of PPAR, CYP80F1, HDH, and H6H.** **a** Extracted-ion chromatograms (EIC) illustrating the in vitro activity of from *Ba*P-PAR, *Mc*PPAR and *Pi*PPAR with phenylpyruvic acid used as substrate. **b** Extracted-ion chromatograms showing the in vivo activity of four *CYP80F1* genes from *A. tanguticus* (*At*CYP80F1), *B. arborea* (*Ba*CYP80F1), *M. caulescens* (*Mc*CYP80F1) and *A. belladonna* (*Ab*CYP80F1) via coexpressing of *Ab*HDH, *At*HDH, *Ba*HDH, *Mc*HDH, *lt*HDH (*I. triloba*), and *In*HDH (*I. nil*) genes, for converting littorine into hyoscyamine in tobacco leaves. Littorine was used as substrates by co-infiltration into tobacco leaves. **c** Extracted-ion chromatograms (EIC) illustrating the in vitro activity of *Mc*H6H and *Lc*H6H with hyoscyamine or anisodamine used as substrate. Source data are provided as a Source Data file.

in validating the functionalities of HS-related genes. Moreover, the overexpression of *TRI* genes in HS-producing species enhances tropine (**6**) production[52–54]. Conversely, our VIGS experiment, aimed at redu-cing the expression of the *TRI* gene in *A. tanguticus*, caused a decrease in tropine (**6**) concentration (Supplementary Fig. 18). These com-plementary experiments reinforce the critical role of the TRI gene in HS biosynthesis, offering consistent evidence. Notably, the con-structed species tree outlining the sampled species supports the findings of previous studies, which indicate that three HS-producing species (representatives of three groups) are phylogenetically distant and do not form a monophyletic clade[55]. This phylogenetic distribu-tion of the HS biosynthetic pathway indicates that it likely originated

via convergent evolution or multiple independent losses following an ancestral origin with syntenic blocks and collinear genes across phy-logenetically unlinked species[5,56]. We annotated syntenic blocks and characterized homologous genes in each HS-synthesis phase across the three non-monophyletic HS-producing species. Moreover, for all 12 HS-synthesis genes, we recognized syntenic blocks and homologous genes, with or without complete genic structure in at least one non-HS-producing species (Figs. 2–4, Supplementary Figs. 19–31). In *HDH*, we identified collinear regions and the complete assembly of homologous genes across Convolvulaceae family species (Supplementary Fig. 31). We also verified its activity via in vivo enzymatic assays (Fig. 6b), indicating that this gene and its function originated prior to the

divergence of the two families. However, suppressing the *HDH* gene did not cause an expected decrease in hyoscyamine (**13**) in *A. tanguticus*, while a previous investigation demonstrated a significant enhancement of hyoscyamine (**13**) production upon *HDH* overexpression[34]. These two experiments indicate that the *HDH* gene may be sufficient but dispensable for hyoscyamine (**13**) biosynthesis, but other yet-to-be-identified genes may influence or complement its function upon its suppression.

These findings support the ancestral origin of the HS biosynthesis pathway prior to lineage diversification of the Solanaceae, with multiple losses similar to other plant traits[5–8]. This concept is further reinforced by the following two facts. First, the gene trees generated according to syntenic genes in HS-producing species are closely aligned with the species trees (Fig. 2b, Supplementary Figs. 19–26, Supplementary Figs. 28, 29, Supplementary Method 1). The genes of each HS-synthesis step across the three HS-producing species do not cluster into a monophyletic clade, as would be found in the case of convergent evolution[44,45] (Supplementary Figs. 19–26). Secondly, crucial sites are conserved across the three HS-producing species, including variations of five crucial sites in the *TRI* gene conserved across various lineages of non-HS-producing species (Fig. 5a, Supplementary Fig. 32). Under convergent evolution, it is unlikely that these similar but crucial sites would be in the same gene to appear in closely related species lacking such a trait[57]. Functional tests support this notion, as mutations to these key sites due to relaxed evolution in non-HS-producing species limited their chemical activity (Fig. 5). An accumulation of mutations in these sites would have resulted in functional inactivity, accelerated pseudogenization, and, ultimately, gene loss in the non-HS-producing species. Our gain-of-function and loss-of-function experiments on *St*TRI and *Ba*TRI via mutual mutation experiments provide strong support for this hypothesis. Upon introduction of mutations found in HS-producing species, we identified the restoration of enzymatic activities in the potato *St*TRI (Fig. 5). This suggests that certain critical sites have undergone limited or terminated activities due to relaxed evolution in non-HS-producing species. Therefore, our systematic comparisons and experimental tests indicate that a single common HS biosynthetic pathway had been established in the ancestral Solanaceae clade, followed by multiple independent losses. These losses likely occurred in response to selection pressure or genetic drift in other closely related lineages throughout diversification and dispersal[3]. The three groups, Datureae, Hyoscyaminaem, and Mandragorinae, that produce HS are distributed across the southern and northern hemispheres[14,58,59], retaining this ancestral pathway under continuously similar selective pressure from herbivorous insects[58,60].

While our investigation did not examine the activities of all identified HS genes across the three HS-producing species, we anticipate that these HS genes will likely be active. This is supported by functional assessments of a limited number of HS genes in other HS-producing species of the same lineages, as reported in another study[22], yielding similar results. Moreover, the VIGS experiments performed on four genes from *A. tanguticus* also offer additional evidence for the vital role of these genes in HS biosynthesis. Furthermore, we found that outside of the three HS-producing species, the *PPAR* gene was also identified in non-HS-producing *Petunia* and *L. chinense*. Similarly, the *HDH* gene was identified in its entirety in two Convolvulaceae species, while it was completely lost in the remaining non-HS-producing species. Both the *PPAR* and *HDH* genes were retained in these non-HS-producing species and exhibited full functionality (Fig. 6). These findings support the notion that HS biosynthesis existed prior to lineage diversification within the Solanaceae family. The random loss of genes in the pathway across certain lineages has resulted in the unequal distribution of HS throughout the Solanaceae. However, syntenic blocks and specific key genes of the HS biosynthesis pathway are retained in some non-HS-producing Solanaceae crops, such as potatoes, tomatoes, tobaccos, and chilies, which are widely cultivated globally. Outside of

establishing an entirely unfamiliar HS biosynthesis route to transfer all related genes into yeasts[18], our findings indicate that it is feasible to engineer the desired HS biosynthesis into non-HS-producing Solanaceae crops. Through genetic modification of homologous genes through site-specific mutations and incorporation of missing genes at collinear regions, it is possible to establish an HS pathway in these crops, obtaining cost-effective and bioactive HS compounds.

## Methods

### Plant materials, library construction, and genome sequencing

Previously, it had been reported that HS could be identified in all species in the three lineages of Solanaceae, Hyoscyaminae, Datureae, and Mandragorinae[9]. To represent each of these lineages, we chose *A. tanguticus*, *B. arborea*, and *M. caulescens* as representatives for each of the three HS-producing lineages, respectively. Early research indicated the presence of scopolamine in *L. chinense* and *L. halimifolium* within the tribe Lycieae[61,62]. However, all subsequent studies focusing on the detection of HS within this tribe failed to uncover their presence in any species or sample[47,63–68]. Consequently, we confirmed *L. chinense* as a representative of the non-HS-producing Lycieae via additional examination (Supplementary Fig. 36). Notably, the genome sequence of another closely related congeneric species, *L. barbarum*, was also reported[69]. Upon comparison of the HS genes of the two species, no fundamental changes were identified. Therefore, we only utilized *L. chinense* to represent the Lycieae lineage. Similarly, we eliminated genome sequences from a back-to-back study[22], which employed two different species, *A. belladonna* and *Datura stramonium* from the HS-producing lineages Hyoscyaminae and Datureae, respectively. This independent study reached independently came to similar conclusion.

High-quality genomic DNA was isolated from fresh young leaves of individual specimens of *A. tanguticus*, *B. arboream*, *M. caulescens*, and *L. chinense* from Menyuan (Qinghai Province, voucher specimens Liujq201904), Chengdu (Sichuan Province, Liujq201905), Guoluo (Qinghai Province, Liujq202203), and Lanzhou (Gansu Province, Liujq202005) China. DNA isolation was conducted utilizing the cetyltrimethylammonium bromide (CTAB) method, and subsequent purification was performed using a QIAGEN DNA purification kit. Each sample underwent short-read genomic library generation with the TruSeq Nano DNA HT Sample Preparation Kit (Illumina USA), following the manufacturer's instructions. Unique barcodes were appended onto the sequences of each sample. These libraries were sequenced on an Illumina NovaSeq 6000 platform, generating 150 bp paired-end reads with an insert size of around 350 bp.

For the Nanopore platform, we chose and purified large fragments using the BluePippin size selection system alongside AMPure beads. Following end-preparation, ligation of sequencing adapters, and tether attachment, all fragments were sequenced using the ONT GridION X5 platform using 6 Nanopore flow cells. For PacBio Sequel II platform, SMRTbell DNA libraries (~20 kb) were generated using the BluePippin size selection system following the PacBio Sequel protocol. Long reads were constructed utilizing the PacBio Sequel II system. Specifically, we obtained reads totaling ~164 Gb from PromethION flow cells on Nanopore for *A. tanguticus*, and ~450 Gb, 477 Gb, and 87 Gb from SMRT cells using the PacBio Sequel II for *L. chinense*, *B. arborea*, and *M. caulescens*, respectively. For transcriptome analyses of each species, RNA-sequencing was conducted using samples from four tissues (leaf, root, secondary root, and stem, each in triplicate) obtained from the same individual for each species. Total RNA was extracted using a RNeasy Plus Mini Kit (Qiagen). The library construction and sequencing were conducted by BGI-Shenzhen Company (Wuhan, China) on the MGI2000 platform using a 2 × 150 bp pair-end model.

### Genome assembly and assessment

A *k*-mer based approach[70] was utilized to estimate genome size. Total clean reads were used to generate the distribution of *k*-mer depth

using $k$-mer size of 17 bp and 23 bp via Jellyfish (v 2.3.0)[71], and GenomeScope (v 2.0)[72] was employed to estimate genome size. De novo assembly of *A. tanguticus* genome using the filtered Nanopore reads was performed using the NextDenovo (v 1.1.1) (https://github.com/Nextomics/NextDenovo.git) assembler program. Sequencing errors were corrected using the NextCorrect module, and an initial assembly was obtained based on the NextGraph module. The genomic contigs were further refined using Racon (v 1.4.10)[73] and Pilon (v 1.23)[74]. Additionally, Nanopore long reads and Illumina short reads were employed twice for error correction based on Racon[73] and Pilon[74] software, respectively. For the assembly of *B. arborea*, *L. chinense*, and *M. caulescens*, the PacBio HiFi raw reads were assembled using hifiasm (v 0.12) (https://github.com/chhylp123/hifiasm) software using default parameters. Further polishing of the genome was performed using Pilon with Illumina data. BUSCO (v 3.0)[75] with 1614 genes from embryophyta odb10, was used to evaluate the completeness and accuracy of the genome assembly. Illumina sequences from DNA and RNA libraries were mapped to evaluate the quality of the assembled genome using Burrows-Wheeler Aligner (BWA) (v 0.7.12-r1039)[76] and HISAT2 (v 2.2.1)[77], respectively.

## Chromosome assignment using Hi-C

The Hi-C (high-throughput chromosome conformation capture) libraries were arranged following a modified procedure to increase the method's efficacy[78]. Specifically, fresh leaves from the sample were exposed to a nuclei isolation buffer containing 1% formaldehyde for crosslinking, followed by grinding the fixed tissue into a powder using liquid nitrogen. The resulting nuclei suspension was isolated and purified. Subsequently, the purified nuclei were digested with 100 units of *Hin*dIII and marked via incubation with biotin-14-dCTP. The ligated DNA was sheared into fragments, and then subjected to blunt-end repair, and A-tailed. Following purification using biotin-streptavidin-mediated pull-down, the Hi-C libraries were quantified and sequenced on the Illumina Hiseq platform (Illumina, San Diego, CA, USA).

In total, 1,064,114,184 bp, 525,935,044 bp, 377,405,779 bp, and 273,678,964 bp of clean paired-end reads were generated from the libraries for *A. tanguticus*, *L. chinense*, *B. arborea*, and *M. caulescens*, respectively. Quality control of Hi-C raw data were performed using Hi-C-Pro[79]. Low-quality sequences (quality scores<20), adapter sequences, and sequences shorter than 30 bp were filtered out using fastp (v 0.20.0)[80] The resulting clean paired-end reads were then mapped to the draft assembled sequence using Bowtie 2 (v 2.3.0)[81] (bowtie2, RRID: SCR_005476) to obtain uniquely mapped paired-end reads.

As a result, a total of 268,827,577 bp, 308,270,700 bp, 258,535,429 bp, and 167,235,469 bp of uniquely mapped pair-end reads were generated *A. tanguticus*, *L. chinense*, *B. arborea*, and *M. caulescens*, respectively. Among these, 80.67%, 77.34%, 88.13%, and 81.25% were valid interaction pairs. Along with the valid Hi-C data, the LACHESIS (v 1.0)[82] (ligating adjacent chromatin enables scaffolding in situ) de novo assembly pipeline was employed to generate chromosome-level scaffolds. A heatmap of the interaction matrix of all pseudochromosomes was charted using a resolution of 100 kb.

## Repeat and non-coding RNA annotation

We utilized two complementary methods (one homology-based and one de novo-based) to expose and classify transposable elements (TEs). A homology-based repeat library was constructed from the known repeat library, Repbase[83] using RepeatMasker (v 4.0.7)[84]. RepeatModeler (v 1.0.11)[85] was employed to construct the de novo-based repeat library. RepeatProteinMask (http://www.repeatmasker.org/) was used for the detection of TEs via comparison to TE protein database. The resulting integrated repeat library was annotated using Tandem Repeats Finder (v 4.04)[86] software. For the prediction of tRNA genes, we utilized the tRNAscan-SE (v 2.0) software package[87].

Additionally, rRNA, miRNA, and snRNA fragments were predicted using INFERNAL (v 1.0)[88] searching against the Rfam database[89].

## Structural and functional annotation of genes

Protein-coding genes within the four Solanaceae genomes were predicted using a comprehensive approach encompassing three distinct procedures (1) de novo predictions, (2) homology-based predictions, and (3) RNA-seq-based predictions. De novo predictions were implemented using three ab initio gene prediction programs: Augustus (v 3.2.3)[90], GENSCAN (v 1.0)[91], and GlimmerHMM (v 3.0.4)[92]. For homology-based predictions, the protein sequences of six species, namely *Arabidopsis thaliana*, *Capsicum annuum*, *Nicotiana tabacum*, *Solanum lycopersicum*, *Solanum pennellii*, and *S. tuberosum*, were aligned to the repeat-masked genome utilizing GeMoMa (v 1.6.1)[93]. We employed two methods to generate RNA-seq-based predictions. One of these involved mapping the RNA-seq data to the genome and assembling the transcripts using Tophat (v 2.1.1)[94] and Cufflinks (v 2.2.1)[95]. The other approach involved the use of Trinity (v 2.8.4)[96] to de novo assemble the RNA-seq data, followed by the employment of PASA (v 2.3.3)[97] to enhance gene structures. To confirm the gene set, all predictions were combined using EVidenceModeler[98] (EVM v 1.1.1) to generate non-redundant gene sets.

Gene functional annotation was performed using BLAST (v 2.3.0)[99] (*E* value ≤ 1e-5) searches against the SwissProt (http://www.uniprot.org/) and NCBI non-redundant (NR) protein databases. Motifs and domains were annotated using InterProScan (v5.36-75.0)[100] to query the InterPro databases, including Pfam, PRINTS, PROSITE, ProDom, and SMART. The Gene Ontology (GO) term[101] for each gene was obtained from the corresponding InterPro descriptions. Additionally, the gene set was mapped to the KEGG[102] pathway database to characterize the most suitable classification for each gene.

## Gene family and phylogenomic analysis

Protein-coding genes derived from a total of 12 species, namely *O. sativa*, *V. vinifera*, *C. sativus*, *M. guttata*, *P. axillaris*, *N. attenuata*, *L. chinense*, *A. tanguticus*, *M. caulescens*, *B. arborea*, *S. tuberosum*, and *S. melongena*, were assessed to characterize gene family groups. Orthologous groups in *A. tanguticus L. chinense*, *B. arborea, M. caulescens*, and the remaining eight other species were recognized using the OrthoMCL (v 2.0.9)[103] program. Expansion and contraction of gene families were identified through a comparison of the differences in cluster size between the ancestor and each species employing CAFÉ (v 4.2)[104] software.

To surmise phylogenetic placements, three procedures were performed. Firstly, all single-copy gene sequences were aligned utilizing MAFFT (v 7.402)[105] using default parameters. Subsequently, a phylogenetic tree of the concatenated sequences from the 12 species was produced using RAxML (v 8.2.12)[106], with the maximum likelihood method and 1000 bootstrap replicates. *O. sativa* was chosen as an outgroup for this phylogenetic tree. Finally, the MCMCtree program within the PAML (v 4.9 h)[107] package was applied to determine the divergence time using the generated phylogenetic tree. The mcmctree parameters were established follows: burn-in of 10,000, sample-number of 100,000, and sample-frequency of 2. Calibration points, obtained from publications and the TimeTree website (http://www.timetree.org), were utilized as normal to constrain the age of the nodes between *V. vinifera* and *O. sativa* (160 Mya). The trees were visualized and edited with the assistance of FigTree (http://tree.bio.ed.ac.uk/software/figtree/).

## Analysis of genome synteny and whole genome duplication

To assess the evolution of the *A. tanguticus, L. chinense, B. arborea*, and *M. caulescens* genomes, we searched for WGD throughout these three Solanaceae genomes. Synteny searches were conducted to characterize syntenic blocks containing more than five collinear genes within a

region utilizing the MCScanX (v 1)[108] package with default parameters. MCScan (Python version v 1.2.7) was used to complement the assessment (https://github.com/tanghaibao/jcvi/wiki/MCscan-(Python-version). The calculation of $K$s values for collinear orthologous gene pairs was accomplished using the Perl script "add_ka_and_ks_to_collinearity.pl" within the MCScanX[108]. Protein sequence anchors were recognized across all possible pair of chromosomes from multiple genomes using BLASTp (v 2.3.0)[99] with an E-value threshold of $\leq$ 1e-5. Ideal matches were indicated in red, while other matches were displayed in blue to assist in distinguishing orthology from paralogy. All dot plots were drawn using WGDI[109] (v 0.4.1). The $K$s value of gene pairs were computed using TBtools (v1.0987)[110] and the simple $K$a/$K$s Calculator (MA) function.

### Identification of HS-synthesis genes and microsynteny analysis

We employed a BLAST search and hidden Markov model (HMM) (v 3.3) to mine for homologs in the genome of three HS-producing species, including *A. tanguticus*, *B. arborea*, and *M. caulescens*, employing the sequence of *A. belladonna*[18,23,32] as a reference. The obtained gene sequences from these two methods were integrated and manually assessed for correction of errors in automatic annotation. To explore the expression of the identified HS-biosynthesis genes throughout the three HS-producing species, we performed RNA-sequencing across four tissues. Low quality reads (with a percentage of low-quality bases over 50% in a read) and unknown bases (> 10%) in the raw reads were removed to retain clean reads. The quality-checked RNA-seq reads were aligned to the four assembled reference genomes using HISAT2 (v 2.2.1)[77] software. The transcripts per kilobase of exon model per million mapped reads (TPM values) were assessed using StringTie (v 2.1.3)[111]. To streamline subsequent analysis, only the final annotated gene models were selected for TPM calculation using the "-e" parameter, lacking any new transcript assembly. Transcriptional network analyses were conducted with WGCNA (v 1.68)[112] (Supplementary Fig. 37). Additional details of the procedures associated with RNA-seq and data analysis are provided in Supplementary Method 2. We evaluated the final HS-synthesis genes in the genome with the genes that have been functionally validated from prior studies.

Genome-wide syntenic blocks were found across 17 selected genomes in this study, following the methodology described by Griesmann et al.[5] (Supplementary Table 28). All versus all Blastp (E value $\leq$ 1e − 5) was performed on the protein sequences of the 17 pairs of annotated gene models, producing a database of protein similarity. Subsequently, the genome sequences of the 17 species were loaded into Multiple Collinearity Scan (MCSCAN, Python version), a package available in the JCVI (v 1.2.7)[113] utility libraries (github.com/tanghaibao/jcvi), responsible for creating a syntenic or collinear block database. The comparisons between gene pairs were conducted using LAST (v 2.32.1)[114] (gitlab.com/mcfrith/lastlast.cbrc.jp). After the removal of hits with reduced scores, the anchors derived from the LAST outputs were clustered into syntenic blocks. Synteny blocks linking each pair of candidate species were performed with parameters "-a -e 1e-5 -s 5". To identify all homologous syntenic blocks associated with HS-synthesis, we employed *A. tanguticus*, *B. arborea*, and *M. caulescens* genomes as references, searching and locating the targeted genes within syntenic blocks with the flanking genes surrounding upstream or downstream 100 kb genomic regions alongside counterparts from other genomes. If a corresponding ortholog was identified in the collinear block from another aligned genome, it was defined as scenario-1 (indicating that synteny supports gene presence, consistent with the genome-wide gene family ortholog prediction). However, if the target orthologous gene was absent from the collinear blocks of the given genome, it was defined as scenario-2 (synteny supports gene absence). After this was finished, we comprehensively assessed the findings of the three genomes and updated the above two scenarios. For species in which no collinear block was found containing the target gene, we manually

assessed the protein database and conducted a search for pseudo-genes in putative regions, defined as scenario-3. Finally, microsynteny plots were generated using the "synteny" function with default parameters.

### Verification of enzyme activity of *TRI*

The coding sequences of *TRI* genes from *B. arborea*, *A. tanguticus*, *M. caulescens*, *L. chinense*, and *S. tuberosum*, as well as *Ba*TRI (one of four tandem duplicated gene), *At*TRI, *Mc*TRI, *Lc*TRI (one of two tandem duplicated gene), and *St*TRI, were amplified and cloned into the protein expression vector pET28a. The five TRI mutant sequences were directly synthesized and cloned into pET28a according to the sequences of their corresponding wild types *St*TRI and *Ba*TRI. All the primers used are outlined in Supplementary Table 29. Subsequently, these constructs were transformed into *Escherichia coli* strain Rosetta (DE3). Following overnight induction with IPTG, the soluble fraction of lysed bacteria was isolated through centrifugation. The recombinant His-tagged TRI were purified using Ni-NTA Agarose (QIAGEN, Germany) according to manufacturer's instructions. To remove imidazole, the samples were dialyzed in ice-cold dialysate (0.05 M potassium phosphate, pH 6.4). The purified recombinant tropinone reductases were assessed using SDS-PAGE electrophoresis.

Enzymatic assays were conducted as follows. Specifically, the substrate reduction activity was assessed by measuring the consumption of NADPH$^+$ spectrophotometrically at 340 nm and 30°C[115]. For each reaction, 20 μg of TRI protein was added to 1 mL of reaction buffer, containing 200 μM NADPH$^+$, 5 mM tropinone (CAS: 532-24-1, Wuhan ChemFaces Biochemical Co., Ltd.) (0.01-16 mM for the determination of $K$m values; the concentration range for each substrate needed to be adjusted according to the results of enzymatic assays), and 0.1 M potassium phosphate at pH 6.4. Data were collected during the initial linear phase of the enzymatic reaction to ascertain the kinetic parameters. The enzyme kinetic constants were computed using non-linear regression of the Michaelis-Menten equation in GraphPad Prism (v 8.3.0)[116]. All assays were conducted in triplicate, and the mean values with standard deviations are reported.

To verify the final product of the catalytic reaction, the samples were subjected to high-resolution mass spectrometry analysis. For the identification of tropine, samples were analyzed using a UPLC-MS (Orbitrap Exploris 120, Thermo Fisher Scientific, Massachusetts) equipped with an ACQUITY UPLC BEH HILIC column (2.1 × 100 mm with 1.7 μm particle size) at a temperature of 35 °C. UPLC separations were performed using a 5 μL injection volume and a flow rate of 0.3 mL/min alongside the gradient illustrated in Supplementary Table 24. MS analyses were performed using electrospray ionization in positive-ion mode and full-scan mode.

### LS enzyme activity assay

Due to the limited production rate of UGT1-produced phenyllactylglucose, it was challenging to isolate a sufficient quantity for the LS enzymatic assay. Therefore, we chose to employ the transient expression method[117] in plants to examine the expected functions of *LS* genes from two HS-producing species, namely *At*LS from *A. tanguticus* and *Ba*LS from *B. arborea*. As a control, we utilized *Ab*LS from the HS-producing *A. belladonna*, belongs to the same lineage as *A. tanguticus* but with an experimentally confirmed function[32]. Additionally, we included *Ab*UGT1 from this species for subsequent chemical reactions. The full-length coding sequences (CDSs) of *YFP* (control), *Ab*UGT1, *Ab*LS, *At*LS, and *Ba*LS were inserted into pEAQ-HT using AgeI and XhoI restriction enzymes (NEB, UK), generating pEAQ-YFP, pEAQ-AbUGT1, pEAQ-AbLS, pEAQ-AtLS, and pEAQ-BaLS, respectively[32]. A list of all primers employed is outlined in Supplementary Table 30.

Subsequently, these constructs were transformed separately into *Agrobacterium* strain GV3101. After culturing the engineered *Agrobacterium* individually, the bacteria were suspended in a buffer

solution, containing 10 mM MgCl$_2$ and 150 mM Acetosyringone (San-gon Biotch, Shanghai, China), and the OD600 of bacterial solution was adjusted to 0.6. The solution was incubated at room temperature for 3 h. The bacterial solution was thoroughly mixed and injected into tobacco leaves. After 96 h, the tobacco leaves were infiltrated with a substrate solution containing 1 mM tropine (CAS:120-29-6, Wuhan ChemFaces Biochemical Co., Ltd.) and 1 mM phenyllactate (CAS: 7326-19-4, Beijing psaitong Biotechnology Co., Ltd.). The infiltrated tobacco leaves were harvested and freeze-dried for metabolite analysis after 24 h. Each experimental group was comprised of six biological repli-cates. The presence of littorine was examined using a Thermo Scien-tific UltiMate 3000 UPLC system coupled to a Thermo Scientific Q Exactive Orbitrap LC-MS instrument, with a Hypersil GOLD C18 column (100 mm × 2.1 mm, 1.9 μm) at a temperature of 40 °C.

## PPAR enzyme activity assay

The coding sequence of *PPARs* from their respective species were amplified via PCR using ApexHF HS DNA Polymerase FL (Accurate Biology, WuHan, China). Subsequently, amplified sequences were inserted into the *E. coli* expression vector pET28a. A complete list of primers used for cloning *PPARs* is outlined in Supplementary Table 31. The recombinant His-tagged PPARs were purified using Ni-NTA Agar-ose (QIAGEN, Germany) based on the manufacturer's instructions. Following purification, the PPARs (20 μg) enzymes were dialyzed in a cold dialysate (50 mM potassium phosphate, pH 8.0), and subse-quently incubated with 3 mM phenylpyruvic acid (CAS: 156-06-9, Chengdu Pengshida Experimental Equipment Co., Ltd.) and 2 mM NADPH (CAS: 2646-71-1, Beijing psaitong Biotechnology Co., Ltd.) at 30 °C for 2 h. A sample of boiled enzyme was used as a negative con-trol. Finally, the reaction was stopped by adding an equal volume of methanol. The catalytic product, phenyllactate, was detected using a Thermo Scientific UltiMate 3000 UPLC system with a Thermo Scien-tific Q Exactive Orbitrap LC-MS instrument with a Symmetry C18 Column.

## CYP80F1 and HDH enzyme activity assay

Due to the absence of a standardized substrate for HDH enzyme activity assays, we utilized transient expression in plants to examine the anticipated functions of the *CYP80F1* and *HDH* genes derived from three HS-producing species (*A. tanguticus*, *B. arborea*, and *M. cau-lescens*). In addition, we validated the functionality of the collinear *CYP80F1* and *HDH* genes from diverse non-HS-producing species (Fig. 4). The CDS of *CYP80F1* and *HDH* were inserted into pEAQ-HT vector using AgeI and XhoI restriction enzymes (NEB, UK) or the OK Clon DNA Ligation technology (Accurate Biology, WuHan, China). We employed the *AbCYP80F1* and *AbHDH* genes from *A. belladonna* as the control[33,34]. All primers used can be found in Supplementary Table 32. These constructs were then independently transformed into *Agro-bacterium* GV3101. The infection experiments of the *CYP80F1* and *HDH* genes in tobacco were completed as outlined above for the *LS* gene. Following the injection of the bacterial solution, the tobacco leaves were infiltrated with a solution containing NADPH (CAS: 2646-71-1, Beijing psaitong Biotechnology Co., Ltd.) and a littorine (CAS: 21956-47-8, Wuhan ChemFaces Biochemical Co., Ltd.) substrate solution. Subsequently, the tobacco leaves infiltrated with substrates were harvested and freeze-dried for metabolite analysis after 24 h. Products was detected by a Thermo Scientific UltiMate 3000 UPLC system with a Thermo Scientific Q Exactive Orbitrap LC-MS instrument using a Hypersil GOLD C18 column (100 mm × 2.1 mm, 1.9 μm) at 40 °C.

## Verification of H6H enzyme activity

As the enzyme activity of the *H6H* gene was already confirmed for *A. tanguticus* and *B. arborea*[47,48], we focused on assessing the H6H enzyme activity in *L. chinense* and *M. caulescens* through in vitro enzymatic assays. The CDS of the *H6H* genes from *M. caulescens* and *L.*

*chinense*, namely *Mc*H6H and *Lc*H6H, were amplified and cloned into the protein expression vector pET28a. All primers used can be located in Supplementary Table 33. Subsequently, these constructs were transformed into *E. coli* strain BL21 star strain (DE3).

All purification procedures were carried out at 0-4°C using Ni-NTA Agarose (QIAGEN, Germany) following the manufacturer' instructions. The purified H6Hs enzymes were dialyzed in cold dialysate (50 mM Tris-HCl, pH 7.4). The enzyme activities of H6Hs were assessed by measuring the formation of products from various substrates. The complete catalytic reaction mixture was composed of 50 mM Tris-HCl buffer (pH 7.4), 0.4 mM FeSO4, 4 mM sodium ascorbate, 1 mM 2-oxo-glutarate, 0.2 mM l-hyoscyamine hydrobromide (CAS: 101-31-5, Wuhan ChemFaces Biochemical Co., Ltd.) or 0.2 mM anisodamine (CAS: 17659-49-3, Wuhan ChemFaces Biochemical Co., Ltd.), 2 mg/mL cata-lase, and the enzyme. The total volume for the enzyme assay was 1 mL[118,119]. Following incubation of the catalytic reaction system at 30 °C for 2 h, NaHCO3 was included to raise the mixture's PH to 9.0, termi-nating the reaction. The mixture was extracted three times using an equal volume of EtOAc and the extracts were pooled and evaporated at 55 °C. Finally, the residue was dissolved in 0.5 ml of methanol and store at 4°C for subsequent use. The alkaloids analysis was performed via Agilent 1260 infinity II High Performance Liquid Chromatography (HPLC) with a ZORBAX Eclipse XDB-C18 column (4.6 mm × 250 mm, 5 μm) at 35 °C.

## VIGS of *TRI*, *LS*, *HDH* and *H6H* genes in *A. tanguticus*

VIGS (Virus-Induced Gene Silencing) was used as a powerful approach for moderately high-throughput characterization of gene function in plants. It was recognized as a particularly effective approach in plants in the Solanaceae family and has been successfully used for silencing root-expressed genes associated with alkaloid biosynthesis[33,120,121]. In this study, we manually removed the seed coat from *A. tanguticus* seeds collected in the wild at high altitudes and placed them on moistened filter paper for germination. Once the radicle emerged, the seedlings were transplanted into nutrient soil (PINDSTRUP) and grown at 24 °C under a 16-h day/8-h night cycle. Approximately one month later, when the second true leaf had emerged, the VIGS test was conducted.

VIGS was performed following a method utilizing the two-component TRV[33] system. The target gene constructs were produced through ligation-independent cloning into the TRV2-LIC vector[122,123]. PCR fragments of the *A. tanguticus PDS*, *TRI*, *LS*, *HDH*, and *H6H* were amplified from the gDNAs from seedlings, using primer pairs descri-bed in Supplemental Table 34. These fragments were directly ligated into the pTRV2 vector, resulting in constructs named pTRV2-*At*PDS, pTRV2-*At*TRI, pTRV2-*At*LS, pTRV2-*At*HDH, and pTRV2-*At*H6H. The recombinant clones were verified using a combination of PCR ver-ification and DNA sequence analysis. The constructs of pTRV1, pTRV2, pTRV2-*At*PDS, pTRV2-*At*LS, pTRV2-*At*TRI, pTRV2-*At*HDH and pTRV2-*At*H6H were transformed into *Agrobacterium tumefaciens* strain GV3101, respectively. After 48 h, single colonies were picked in 1 ml of YEP medium supplemented with 50 mg/L rifampicin (Rif), 50 mg/L kanamycin (Kan) and 10 mg/L gentamycin (Gen) sulfate at 28 °C for overnight activation. After detected by PCR, the bacteria were inocu-lated into 100 mL (Kan+Rif+Gen) YEP liquid medium with a con-centration of 2% and cultured overnight at 28 °C.

On the next day, the activated bacteria were centrifuged at 4500 rpm/min for 10 min. The supernatant was discarded, and the bac-terial were resuspended in buffer to an OD$_{600}$ = 1.0. The suspension was then allowed to stand for 3 to 5 h at room temperature. Prior to injection, the pTRV1 was mixed with all pTRV2 recombinant vectors of *A. tumefa-ciens* strain GV3101 at a ratio of 1:1. Finally, the silencing experiments were conducted by infiltrating *Agrobacterium* cultures into the cotyle-dons of *A. tanguticus* seedlings using two methods. The first method involved leaf injection, in which a sterilized needle was used to draw up the bacterial solution and inject it into the dorsal veins of the true leaves.

The second method involved root irrigation, in which a sterilized pipette was used to draw up 5 mL of bacterial solution, which was then inserted into the soil near the roots. Infiltration was repeated every 7 days, and each method was conducted three times. Following infiltration, the plants were cultured in darkness for 48 h and then transferred to a growth room maintained at 24 °C with a 16-h photoperiod. After 28 days, the roots of the infected plants were obtained for gene expression analysis. Six independent expression down-regulated lines were used for subsequent alkaloid measurement. Tissue samples used for alkaloid extractions were gathered from the infected plants after 45 days.

An appropriate amount of infected plant tissue from *A. tanguticus* was ground in liquid nitrogen. Total RNA was extracted using a Plant RNA extraction kit-V1.5 (BIOFIT, China). First-strand cDNAs were synthesized using Hifair® III 1st Strand cDNA synthesis SuperMix for qPCR (Cat NO.11141ES60, YEASEN, Shanghai, China). The *PGK* gene was utilized as the internal reference to assess the expression of each target gene. All primer sequences can be identified in Supplementary Table 34. For qRT-PCR reactions, the Hieff UNICON® Universal Blue qPCR SYBR Green Master Mix (Cat NO.11184ES08, YEASEN, Shanghai, China) was used.

The extraction and assay of alkaloids from infected plants were performed following methods. Firstly, metabolites were isolated from each line. The identification and quantification of tropane alkaloids were achieved using an LC-MS/MS system (LC, Shimadzu LC30AD; MS, QTRAP 6500+) coupled to a Thermo Hypersil Gold analytical column (100 mm × 2.1 mm, 1.9 μm) at 40 °C. Quantitative analyses were performed using electrospray ionization in positive ion mode with multiple reaction monitoring. A series of calibration standards and blanks were assessed alongside each sample set.

## Metabolite extraction and detection
Metabolite extraction was conducted in the following method. Initially, the plant samples were lyophilized and finely pulverized into a powder. Metabolites were extracted from these frozen powders with 1 mL extraction buffer containing 20% methanol and 0.1% formic acid per 25 mg of sample powder. Following centrifugation at $13,000 \times g$ for 5 min, the resulting supernatant was compiled for UPLC-MS analysis. Tropine (CAS:120-29-6, Wuhan ChemFaces Biochemical Co., Ltd.), littorine (CAS: 21956-47-8, Wuhan ChemFaces Biochemical Co., Ltd.), phenyllactate (CAS: 7326-19-4, Beijing psaitong Biotechnology Co., Ltd.), and hyoscyamine (CAS: 101-31-5, Wuhan ChemFaces Biochemical Co., Ltd.) were examined using UPLC-MS. The instrument parameters were as follows: a source voltage of 3.0 kV, a capillary temperature of 350 °C, an S-lens RF level of 50, an auxiliary gas heater temperature of 350 °C, a flow rate of 0.5 mL/minute and a mobile phase composed of 0.1% formic acid in water (pump A) and acetonitrile (pump B).

For the detection of anisodamine (CAS: 17659-49-3, Wuhan ChemFaces Biochemical Co., Ltd.) and scopolamine (CAS: 51-34-3, Wuhan ChemFaces Biochemical Co., Ltd.) an HPLC system coupled to a ZORBAX Eclipse XDB-C18 column was applied. The mobile phases contained 0.02% formic acid. The elution procedures for littorine, phenyllactate, hyoscyamine, anisodamine and scopolamine can be found in Supplementary Tables 35 and 36. For the detection of alkaloids in *L. chinense*, different tissues used for measurement were initially ground into coarse powder and dried at 60 °C for 8 h. To extract the metabolites from this powder, an extraction buffer consisting of 0.2 g of $NaCl_2$, 8.6 ml of ethanol, 1 ml of water, and 0.4 ml of formic acid was added to 1 g of sample powder. After undergoing ultrasound treatment for 5 min, the resulting supernatant was collected for HPLC-MS analysis using a Wondasil C18 column (150 mm × 4.6 mm, 5 μm).

## Inferences of ancestral states for key sites of HS-synthesis genes
FastML (v 3.11)[124] software was applied to perform Maximum Likelihood ancestral state inference on amino acid sequences of TRI alignment across the 17 Solanaceae species. An identical amino acid frequency, alpha, and amino acid substitution model used in the sequence simulation was employed for the inference. The state with the highest probability was retained as the inferred ancestral state within the Maximum Likelihood framework.

## Reporting summary
Further information on research design is available in the Nature Portfolio Reporting Summary linked to this article.

## Data availability
All genomic sequencing data and transcriptomic raw data have been deposited in the NCBI Sequence Read Archive (SRA) under BioProject accession PRJNA765943 for *Anisodus tanguticus*, PRJNA765960 for *Brugmansia arborea*, PRJNA765963 for *Lycium chinense* and PRJNA903289 for *Mandragora caulescens*. The four species genome assemblies and annotations are also available at Figshare: *Anisodus tanguticus* [https://figshare.com/s/ecd289d0edea48c839b0], *Brugmansia arborea* [https://figshare.com/s/91c3ad78a30fed3fcdb4], *Lycium chinense* [https://figshare.com/s/9b601f69085726ab3b2a], *Mandragora caulescens* [https://figshare.com/s/b95148b8ae0d72fb7f0f]. Source data are provided with this paper.

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

## Acknowledgements

This work was supported equally by the Second Tibetan Plateau Scientific Expedition and Research (STEP) program (2019QZKK0502) and the Strategic Priority Research Program of the Chinese Academy of Sciences (XDB31000000) and the Key Science & Technology Project of Gansu Province (22ZD6NA007) and the Fundamental Research Funds for the Central Universities (lzujbky-2022-ey07 to Y.Y.) and the Young Talent Development Project of State Key Laboratory of Herbage Improvement and Grassland Agro-ecosystems (No. 2021+02 to Y.Y.) and International Collaboration 111 Program (BP0719040). We would like to thank the support for computational work from Supercomputing Center of the Lanzhou University.

## Author contributions

J.L. designed and led the project. J.L., J.Y., J.W., C.Z., and X.F. collected samples in the field. J.Y., Y.W., J.M. and Y. Yang performed the assembly, repeat and gene annotations of the three genomes. J.Y. performed the polyploidization analysis. J.Y., Y.W., and J.M. carried out the gene family analysis and the phylogenomic analysis. Y.W. and J.Y. performed the microsynteny analysis. P.Z., Y. Yao, L.Z., and Y.M. carried out the enzyme assays. J.Y., J.L., Y.W. and Y. Yang. wrote and edited most of the manuscript. All of the authors read and approved the final manuscript.

## Competing interests

The authors declare no competing interests.
