## [Peer Review File · Nature Communications]

Multiple independent losses of the biosynthetic pathway for two tropane alkaloids in the Solanaceae familyReviewers' Comments:

Reviewer #1:

Remarks to the Author:

The work entitled "Multiple biosynthesis losses of two specific tropane alkaloids in the Solanaceae" by Yang et al. is describing the sequencing at chromosome level of genomes from four plants in family Solanaceae. The aim of this work is to address the evolution of the tropane alkaloids hyoscyamine and scopolamine biosynthetic pathway (authors refer to it as HS). The selection of the three HS producing plants *Anisodus tanguticus*, *Brugmansia arborea* and *Mandragora caulescens* is reasonable and reflect three different Solanaceae tribes with HS producing plants. The phylogenomic analysis that helped to reach some conclusions about the pathway evolution in family Solanaceae assuming that syntenic relationships are connected to conserved activity without including some biochemistry data.

The authors used as outgroup the genomes of two *Petunia* species from the Solanaceae *Petunioideae* subfamily. The authors include a mutational analysis for the catalytic activity of TRI enzyme which is well done however it does not feel connected with major content of the paper which is the genome sequencing of four plants and the evolution of HS pathway in Solanaceae family.

The introduction in HS metabolism is not easy to follow since the figure showing the pathway is placed in supplementary information and not in main manuscript though it feels a bit like a copy from the ref 9 and maybe it should be redesigned.

Reading the manuscript, more questions raised, and I have some concerns regarding the work design. Biosynthesis of hyoscyamine and scopolamine is consisted of the biosynthesis of tropane ring and PLA glucosides. Littorine synthase (a serine carboxypeptidase (SCP)-like acyltransferase) is the entry step in HS pathway. Tropane alkaloids are found in different species across family Solanaceae and Convolvulaceae (authors should include the TRI as part of tropane ring biosynthesis in Figure 4).

Tropane alkaloids calystegines have been isolated from various *Ipomoea* species and since the genomes of various of *Ipomoeae* species (eg morning glory and sweet potato) are available may serve as better outgroup. At the end, the authors used the genomes of *Ipomoeae* to establish syntenic relationships to HDH genes of the three HS-producing species (*A. tanguticus*, *M. caulescens*, *B. arborea*) in Supplementary Figure 26.

The authors excluded from their comparative genomic analysis the available genomes of HS producing plants *Lycium barbarum* (Lyciinae) and *Datura stramonium* (Datureae) that might offer more insights regarding the evolution of HS pathway in Solanaceae. The authors claim that the wolfberry species *Lycium chinense* representing tribe Lyciinae is a non HS producing species. Hyoscyamine presence in *Lycium chinense* has been reported in several books (Sun, et al., Brief Handbook of Natural Active Compounds, Medicinal Science and Technology Press of China, Beijing, (1998); Ou, et al., Brief Handbook of Components of Traditional Chinese Medicines, The People's Medical Publishing House, Beijing, (2003); Chang, et al., Dictionary of Chemistry, Science Press, Beijing, (2008); Chen, Liu, et al., Determination of Effective Components in Traditional Chinese medicines, People's Medical Publishing House, Beijing, (2009)) according to <http://www.knapsackfamily.com/KNAPSAck/> database. Therefore, the claim of the HS independent loss in *Lycium chinense* is rather problematic and it cannot be representative of tribe Lyciinae as it is shown in Figure 4.

The authors explain the pseudogenization through changes in intron and exon structures and exon losses. The conclusions about the HS pathway evolution are mainly based on the observed syntenic relationships. The observed synteny erosion does not mean necessarily that the gene loss or the metabolic gene translocation are always associated with the loss of a metabolic trait. The authors show that there is no syntenic relationship among the PPAR genes in the *A. tanguticus*, *M. caulescens*, *B. arborea* while there are no supporting biochemical data. This grows more questions if the syntenic genes PPAR in *Petunia*, CYP80F1 in *Petunia*, HDH in *Ipomoea*, and H6H across many species (Figure 4), have enzymatic activities in HS pathway. The authors have explored the TRI enzymatic activities providing important insights about tropane ring biosynthesis but tropane ring formation seems to be present in most of species in order Solanales.

My suggestion is that the authors should enrich their work with biochemical data on more key enzymatic steps based on their bioinformatic analysis.

Reviewer #2:

Remarks to the Author:

The manuscript entitled „Multiple biosynthesis losses of two specific tropane alkaloids in the Solanaceae” tries to shed light on the biosynthesis of tropane alkaloids specifically hyoscyamine and scopolamine. To this aim, the authors sample within the Solanaceae and sequence three genomes of species that produce these tropane alkaloids (*Anisodus tanguticus*, *Brugmansia arborea*, *Mandragora caulescens*) and one (*Lycium chinense*) that does not. They then compare the genomes to existing genome assemblies of Solanaceae species and determine that the pathways generally seem to be there. Indeed, cloning a potato TRI homolog (i.e. a non-producer) did not show activity. In total the authors produce some interesting evidence but there might be issues with the genome analysis see below.

The main interesting findings were

- 1) chromosome level assemblies of all four species
- 2) An identification of likely whole genome duplication events
- 3) Demonstration that no activity could be found for potato TRI and identification of two crucial sites for TRI synthesis
- 4) Demonstration that the tropane pathway was likely ancestral and was subsequently lost in some species.

Comments

1) One finding are the WGD events. Whilst the number of chromosomes make these very likely, there are some issues which should be checked not only for the sake of the WGD analysis, but to obtain a decent genome annotation and to investigate these genomes in the future as a resource.

Here are the observations:

- a) *L. chinense* is the most heterozygous genome at ca 0.9% het estimated in S-. Fig 2
- b) root mapping rates for RNAseq data in *Brugmansia* is low S table 8
- c) BUSCO and some annotation statistics

Species duplicated BUSCO S Table 10 Total genes RVM S Table 11ff Annotated NR S Table 17

Anisodus tanguticus	28.5 %	46606	92.1%
Brugmansia arborea	3.9%	32347	92.2%
Lycium chinense	12.3%	54946	79%
Mandragora caulescens	14.1%	29193	93.8%

Now *Mandragora* and *Anisodus* should have seen a WGD however the numbers for BUSCO duplications are relatively low, but this could be due to an older WGD event which can be discussed and should be looked at. However, the high value of 12.3 % duplicated BUSCO in *Lycium* can not be explained, unless these are unremoved haplotypes (see het rate above) as *Lycium* is supposed not to have a WGD. This is somewhat supported by gene number as well (too high), but given that *Lycium* has a much lower NR similarity rate (same for Swissport) in addition there seem to be an overannotation of genes potentially.

The low mapping rates for *Brugmansia* data however are not reflected in the BUSCO or annotation rates, here one could only speculate for an issue in RNAseq.

In general as also shown in S Table 15, at the very least the gene numbers for *L.chinense* are too high. For *A. tang.* And *M. caul.* (WGD and 24 chromosomes !) it would make sense to investigate why the gene numbers are not even close to 2x that of potato and tomato and in the case of *M.caul* even less than 1x potentially indicating an underannotation.

Generally gene structural annotation is not too easy and many ab-initio tools overannotate. But functional sub-annotation or comparison with references can also help improve this.

2) Data must be made available

3) As the authors identified potential potato TRI residues that seem to be important for activity would it not make sense to consider the activity of the potato TRI mutant A109V and L167V as well, i.e.

Creating a gain of function mutant? Even though the loss of function study is already interesting. (But we know they are necessary for function but are these two residues sufficient as well?)

Minor and typos

Page 4 line 100: "Lycium chinense, 102 respectively (Table 1, Supplementary Fig. 1," that is S Fig 2

Page 5 line 133: in the three gene sets: should likely be in the four gene sets

Page 14 line 381 Pacbio raw reads = probably PacBio HIFI reads are meant (i.e. preprocessed data)

Page 15 line 390 HiC data is given as approx. 1Gb, 525Mb 377MB and 273MB I think the numbers are reads numbers (HiC data in this volume would not be enough to scaffold genomes)

Page 18 line 476 "Transcripts were assembled and quantified using Stringtie" Even though Stringtie produces high quality annotations, I would still use the final genome annotation results and not interim Stringtie results. But take the potential issues from above.

Unfortunately checking is not possible as data is not available.

Brugmansia was shown (? Or assembled) to 13 chromosomes. Is there evidence for the deviation from the base number. There seems to be little cytological data but the data available seems to indicate Brugmansia arborea (previously D. arborea) had n=12 chromosomes see

<https://doi.org/10.1080/00087114.1968.10796312>

That said it could be earlier species misidentifications from the Datura genus as well but needs to be discussed + checked at least.

S figure 9 : Boxes seem to be drawn arbitrarily, compare a and b. First box in b is also visible in a. This should be done more exhaustively

Throughout the manuscript leaded = led

Reviewer #3:

Remarks to the Author:

The present manuscript describes a part of the evolutionary history of tropane alkaloid (TA) biosynthesis in the Solanaceae family. While focusing on hyoscyamine and scopolamine synthesis, the authors sequenced the genome of three species producing these two TAs and one non-producing species. By combining this new dataset to 11 previously sequenced genomes, authors analyzed the synteny of the hyoscyamine and scopolamine biosynthetic pathway and established the high collinearity between the corresponding genes. This allowed them to propose an ancestral origin of this TA biosynthetic pathway before lineage diversification, accompanied by multiple losses (eg: exon losses) leading to pseudogenization and loss of hyoscyamine and scopolamine biosynthesis capacity. Such a notion was also reinforced by the characterization of the mutations in the sequence of the Tropinone reductase I that did not let to activity loss in some hyoscyamine and scopolamine nonproducing plant species. In sum, this is an informative describing work. In addition, experiments have been well conducted, the results are convincing, and the manuscript is well written. Comments and Concerns are listed below:

- Unify the term specialized and secondary metabolites
- Both DNA and RNA extraction protocols should be described. HiC library preparation should be detailed as well.
- Long-read library preparation should be described prior giving sequencing results
- Cufflinks is considered depreciated now as there is no maintenance for 10 years. other tools should be preferred (StringTie, Trinity...)
- Was trinity run on its own or using the genome-guided option?
- Please describe RNA extraction method

- Please precise if StringTie assembly was performed individually and merged (using stringtie -merge) or using all RNAseq data together.
- Be consistent in the tools you used: was genome-guided transcriptome assembly performed using cufflinks (L424) or StringTie (L476)?
- LL114-117: four genomes are listed but only three %
- LL359-361 "We obtained polymerase reads totaling ~164Gb, ~450Gb, ~477Gb and ~87Gb from PromethION flow cells on Nanopore and SMRT cells on PacBio Sequel II for *A. tanguticus*, *L. chinense*, *B. arborea* and *M. caulescens* respectively." You should rephrase as it is unclear which species has been sequenced with ONT and PacBio.
- L364: please precise NovaSeq 6000
- L371: please be consistent when precisng the PacBio Sequel you used.
- All versions of software and parameters should be mentioned.
- The authors have mutated two key residues of BaTRI leading to a dramatic loss of activity. Did they consider performing the reverse mutations in StTRI? This would be helpful to determine the role of this mutation in the loss of activity. Indeed, StTRI and LcTRI bear the same residues but different activities.

Reviewer #1

The work entitled “Multiple biosynthesis losses of two specific tropane alkaloids in the Solanaceae” by Yang et al. is describing the sequencing at chromosome level of genomes from four plants in family Solanaceae. The aim of this work is to address the evolution of the tropane alkaloids hyoscyamine and scopolamine biosynthetic pathway (authors refer to it as HS). The selection of the three HS producing plants *Anisodus tanguticus*, *Brugmansia arborea* and *Mandragora caulescens* is reasonable and reflect three different Solanaceae tribes with HS producing plants. The phylogenomic analysis that helped to reach some conclusions about the pathway evolution in family Solanaceae assuming that syntenic relationships are connected to conserved activity without including some biochemistry data.

1. The authors used as outgroup the genomes of two *Petunia* species from the Solanaceae Petunioideae subfamily. The authors include a mutational analysis for the catalytic activity of TRI enzyme which is well done however it does not feel connected with major content of the paper which is the genome sequencing of four plants and the evolution of HS pathway in Solanaceae family.

Reply: In the revised manuscript, we included *Ipomoea nil* and *I. triloba* from the Convolvulaceae family as outgroup. Our analyses of genomes from three HS lineages and non-HS lineages showed that certain HS genes lost their functionality in the non-HS lineages. Specifically, the **TRI gene, which produces tropine, a crucial intermediate metabolite in the HS pathway¹ was examined.** To validate our bioinformatics findings, we conducted functional verification experiments on the TRI gene in various Solanaceae species, including both HS and non-HS species. The results demonstrated that the TRI genes either lose their function or experienced weakened functionality in the non-HS species due to the relaxed evolution (Fig. 5). Furthermore, we performed gain-of-function experiments on the TRI gene from one non-HS species, potato, which resulted in the expected increase in enzyme activity (see Fig. 5). Therefore, the TRI experimental results consistently support the major phylogenetic analyses conducted on the genomes.

2. The introduction in HS metabolism is not easy to follow since the figure showing the pathway is placed in supplementary information and not in main manuscript though it feels a bit like a copy from the ref 9 and maybe it should be redesigned.

Reply: Done. Figure 2a shows the overview of HS biosynthetic pathway in the revised manuscript.

3. Reading the manuscript, more questions raised, and I have some concerns regarding the work design. Biosynthesis of hyoscyamine and scopolamine is consisted of the biosynthesis of tropane ring and PLA glucosides. Littorine synthase (a serine carboxypeptidase (SCP)-like acyltransferase) is the entry step in HS pathway. Tropane alkaloids are found in different species across family Solanaceae and Convolvulaceae (authors should include the TRI as part of tropane ring biosynthesis in Figure 4). Tropane alkaloids calystegines have been isolated from various *Ipomoea* species and since the genomes of various of *Ipomoeae* species (eg morning glory and sweet potato) are available may serve as better outgroup.

Reply: Thank you for your excellent suggestion, which enhances the overall logic and integration of the manuscript. In response to this, we have included *Ipomoea nil* and *I. triloba* species as outgroup species in the revised manuscript. Regarding the TRI gene, it holds significant importance in the formation of the tropane ring. Given that the product of the TRI gene, tropine, serves as a crucial intermediate metabolite in the biosynthesis of medicinal hyoscyamines, we have placed the

TRI gene within the section dedicated to medicinal hyoscyamine biosynthesis in Figure 4.

4. At the end, the authors used the genomes of *Ipomoea* to establish syntenic relationships to HDH genes of the three HS-producing species (*A. tanguticus*, *M. caulescens*, *B. arborea*) in Supplementary Figure 31. The authors excluded from their comparative genomic analysis the available genomes of HS producing plants *Lycium barbarum* (Lyciinae) and *Datura stramonium* (Datureae) that might offer more insights regarding the evolution of HS pathway in Solanaceae.

Reply: Both *Lycium barbarum* and *L. chinese* are closely related species, belonging to the same non-HS lineage discussed in this study. While preparing our manuscript, the genome of *L. barbarum* was published. We thoroughly examined all HS-related genes in both species and found no sequence difference. Therefore, it was deemed unnecessary to repeat the analyses of these genes. We cited the published paper acknowledging the availability of the of the *L. barbarum* genome and explained that we opted to utilize our own genome in order to maintain conciseness in the figures and results. Additionally, we highlighted that the quality of our final genome assembly surpasses that of *Lycium barbarum*.

Similarly, *Brugmansia arborea* and *Datura stramonium* represent an isolated monophyletic HS lineage. In our analyses, we solely utilized *Brugmansia arborea* as a representative of this HS lineage. Another research study, conducted independently but in close proximity to ours, improved upon the previous genome assembly of *Datura stramonium*²⁻³ and enabled us to compare the syntenies of the HS genes²⁻³.

Both our manuscript and theirs were submitted to Nature Communications around the same time in 2022, without prior knowledge of each other's results or submissions. While we were in the process of conducting additional experiments and refining our work, their results were published in Nature Communications.

Due to the timing and overlap, we refrained from utilizing their results for comparisons. However, we conducted a synteny analysis of the 12 HS genes between the two species, which we present here for your examination. The analysis revealed high synteny among all 12 HS genes (see FigR.1), and our further comparisons indicated that the findings align with the conclusions based on *Brugmansia arborea* in the current revised manuscript. Importantly, the inclusion of *Datura stramonium* did not alter the overall conclusion, and therefore, we opted to focus on a single species for each HS lineage in order to maintain conciseness in the manuscript.

FigR. 1 Microsynteny analysis of 12 HS biosynthetic pathway genes between *B. arborea* and *D. stramonium* species in Datureae. a-k, The Microsynteny analysis of *PMT*, *MPO*, *PYKS*, *CYP82M3*, *TRI*, *AT4*, *PPAR*, *UGT1*, *LS*, *CYP80F1*, *HDH* and *H6H* genes, respectively.

4. The authors claim that the wolfberry species *Lycium chinense* representing tribe Lyciinae is a non HS producing species. Hyoscyamine presence in *Lycium chinense* has been reported in several books (Sun, et al., Brief Handbook of Natural Active Compounds, Medicinal Science and Technology Press of China, Beijing, (1998); Ou, et al., Brief Handbook of Components of Traditional Chinese Medicines, The People's Medical Publishing House, Beijing, (2003); Chang, et al., Dictionary of Chemistry, Science Press, Beijing, (2008); Chen, Liu, et al., Determination of Effective Components in Traditional Chinese medicines, People's Medical Publishing House, Beijing, (2009)) according to <http://www.knapsackfamily.com/KNAPsAcK/> database. Therefore, the claim of the HS independent loss in *Lycium chinense* is rather problematic and it cannot be representative of tribe Lyciinae as it is shown in Figure 4.

Reply: Thank you for bringing up this concern. We apologize for not clarifying this issue in the previous version of the manuscript (which has now been addressed in the materials and methods section). This dispute has indeed been clearly addressed in China, as wolfberries are widely consumed as food in northwest China, making this issue critical for wolfberry markets and food safety. Numerous repeated studies and market samplings have been conducted to examine hyoscyamine sulfate (HS) in wolfberries of the *Lycium* genus. HS is prohibited for use as food in China due to its analeptic and delusive effects. However, there may have been some misunderstanding regarding the presence of HS in the Lyciinae subtribe. This could be attributed to the confusion between scopoletin and scopolamine, as well as potential errors in early research studies, such as the use of incorrect materials or the mishandling of experiments or instruments.

Furthermore, *Lycium chinense* has lost three vital genes downstream of the HS synthesis pathway (*LS*, *CYP80F1*, and *HDH*) (see Fig. 4). A back-to-back research study³ failed to detect HS in *Lycium chinense* and observed the loss of several HS-specific genes, including *LS*, in this species. HS cannot be produced in the absence of these HS-specific genes. We have provided a list of major references to support this claim, and our experimental analyses are included in the supplementary file of the revised manuscript.

The study of tropane alkaloids present in wolfberries dates back to 1984 when Drost-Karbowska⁴ reported the presence of both atropine and scopolamine in *Lycium halimifolium*. Similarly, in 1989, Harsh⁵ reported the presence of atropine and scopolamine in fresh *Lycium barbarum* in the fruit, leaf, and cultivated tissues. These reports had a significant impact on the wolfberry food markets in China. Wolfberries have been used as a special food for over 2000 years in China, with no reports of analeptic and delusive effects⁶. However, due to increasing concerns about the safety of Chinese medicinal food⁷ and the wide-ranging value of wolfberries, there has been ongoing discussion and exploration of the types of tropane alkaloids and their contents in the *Lycium* genus.

In 2006, Austrian scholar Bauer⁸ analyzed eight samples of *Lycium barbarum* from China and Thailand using HPLC-MS. The results clearly rejected the presence of scopolamine in this species. In 2011, Yao et al.⁹ examined tropane alkaloids in 17 samples of wolfberry from different regions and species in China, including both *Lycium barbarum* and *L. chinense*, using an LC-MS method. They also failed to detect scopolamine in any of the species or samples. In 2016, Kokotkiewicz¹⁰ used thin-layer chromatography to analyze the fruits, roots, stems, and leaves of three varieties of *Lycium barbarum* and found no detectable scopolamine. In 2020, Zhao and Shi¹¹ developed a new method using UPLC-MS for the simultaneous quantification of three tropane alkaloids in wolfberries. The results from 30 different regions of *Lycium barbarum* showed no anisodamine and scopolamine in any of the samples¹².

To address your request, we further conducted measurements of the presence and content of tropane alkaloids in wolfberries. We collected *L. chinense* seedlings from Lanzhou and grew them in the laboratory. After over a month, we collected root, stem, and leaf tissues. We also purchased wolfberry fruit from local markets, including *Lycium barbarum*. We detected tropane alkaloids using HPLC-MS (see FigR. 2). Furthermore, we failed to detect hyoscyamine and scopolamine in any of the wolfberry tissues (see FigR. 3-5). We performed three biological replicates for each tissue sample. All of our results indicated the absence of hyoscyamine and scopolamine. **In fact, the genome sequences of both wolfberry species lack three key genes, *LS*, *CYP80F1*, and *HDH*, in the HS pathway, making it unlikely for *Lycium* plants to produce hyoscyamine and scopolamine.**

FigR. 2 The morphologies of wolfberry. a, b, overall morphology of wolfberry. c root, d, stem, e, leaves, f, fruit.

Fig. 3 HPLC-MS analyses of hyoscyamine and scopolamine standard. a, HPLC analyses of two standards. b, MS spectra of two standards.

FigR. 4 MS spectra for determination of hyoscyamine in various tissues of *L. chinense*. a, MS spectra of hyoscyamine standard. b, MS spectra in the root of *L. chinense*. c, MS spectra in the stem *L. chinense*. d, MS spectra in the leaf *L. chinense*. e, MS spectra in the fruit of *L. chinense*.

FigR. 5 MS spectra for determination of scopolamine in various tissues of *L. chinense*. a, MS spectra of scopolamine standard. b, MS spectra of *L. chinense* root. c, MS spectra of *L. chinense* stem. d, MS spectra of *L. chinense* leaf. e, MS spectra of *L. chinense* fruit.

5. The authors explain the pseudogenization through changes in intron and exon structures and exon losses. The conclusions about the HS pathway evolution are mainly based on the observed syntenic relationships. The observed synteny erosion does not mean necessarily that the gene loss or the metabolic gene translocation are always associated with the loss of a metabolic trait.

Reply: Yes. You are right that synteny erosion does not mean the necessary gene loss. We mean that some genes, for example, *LS*, *CYP80F1* and *HDH* had lost in the non-HS species due to this reason. We changed the related statements in the revised manuscript.

6. The authors show that there is no syntenic relationship among the *PPAR* genes in the *A. tanguticus*, *M. caulescens*, *B. arborea* while there are no supporting biochemical data.

Reply: Thank you for bringing this to our attention. We have revised our statements accordingly. The microsynteny analyses of *PPAR* did not reveal direct syntenic blocks between *B. arborea* and *A. tanguticus* or *M. caulescens*, nor between these species and non-HS-producing species such as *Petunia inflata*, *P. axillaris*, and *Lycium chinense* (see Supplementary Figure 30a). However, we confirmed candidate ortholog genes of these species through genome-wide gene family analysis (see Supplementary Figure 30b). In a recent study on PPARs from tea plants, the transformation of phenylpyruvic acid into phenyllactic acid was also observed¹³. Based on this finding, we hypothesized that this enzyme is widely distributed across the Solanaceae family and other groups. To investigate further, we conducted *in vitro* enzymatic assays to characterize the function of PPARs from *B. arborea*, *M. caulescens*, and *P. inflata*. Consistent with our hypothesis, all tested PPARs were able to convert phenylpyruvic acid into phenyllactate (see Fig. 6a). This suggests that the *PPAR* gene is unlikely to be a limiting factor in the biosynthesis of phenyllactylglucose and may have shared functions in both the HS pathway and other pathways.

7. This grows more questions if the syntenic genes *PPAR* in *Petunia*, *CYP80F1* in *Petunia*, *HDH* in *Ipomoea*, and *H6H* across many species (Figure 4), have enzymatic activities in HS pathway.

Reply:

As previously described for *PPAR*, we cloned and transiently expressed the *CYP80F1* homologs in tobacco leaves to assess their enzymatic function. However, the results revealed that the *CYP80F1* gene of *P. inflata* (*PiCYP80F1*) lacked a complete functional domain compared to other HS-producing species (see Supplementary Figure 34). Therefore, we conducted relative enzyme activity assays for four *CYP80F1* homologs in *A. tanguticus* (*AtCYP80F1*), *B. arborea* (*BaCYP80F1*), *M. caulescens* (*McCYP80F1*), and *A. belladonna* (*AbCYP80F1*) by transiently expressing them in tobacco leaves along with *AbHDDH*, *AtHDDH*, *BaHDDH*, *McHDDH*, *ItHDDH* (*I. triloba*), and *InHDDH* (*I. nil*), which convert littorine into hyoscyamine. After infiltrating littorine, we detected both littorine and hyoscyamine in tobacco leaves expressing the constructs (see Fig. 6b). This suggests that all the tested enzymes were active.

Bioinformatics analysis and *in vitro* enzyme activity assays demonstrated that the *H6H* gene of *M. caulescens* (*McH6H*) and *L. chinense* (*LcH6H*) exhibited enzyme activity (see Fig. 6c, Supplementary Figure 35). Previous studies have reported that the *H6H* gene of *A. tanguticus* and *B. arborea* catalyzed the two-step reaction of hyoscyamine production to yield scopolamine¹⁴⁻¹⁵. Additionally, to further confirm the roles of the *HDH* and *H6H* genes in HS-producing species, we employed VIGS approaches for in-planta validation in *A. tanguticus*. **The results indicated that suppressing the expression of *H6H* genes significantly decreased the contents of the corresponding products anisodamine and scopolamine in the roots of *A. tanguticus* (see Supplementary Figure 18).** However, suppressing *HDH* gene did not decrease the expected production of hyoscyamine in *A. tanguticus*. Notably, a previous investigation demonstrated a significant increase in hyoscyamine production upon overexpression of *HDH*¹⁶. This observation suggests that the *HDH* gene serves as a sufficient but dispensable factor for hyoscyamine biosynthesis in this species. Consequently, it is reasonable to propose the existence of other yet-to-be-identified genes that potentially complement the activity of *HDH*. These findings lead us to conclude that the identified genes are necessary for HS biosynthesis in plants. Collectively, the HS-synthesis pathways likely originated in the common ancestor of all Solanaceae lineages, and the loss of genes in non-HS-producing tribes contributed to the limited distribution of HS in Solanaceae.

8. The authors have explored the *TRI* enzymatic activities providing important insights about tropane ring biosynthesis but tropane ring formation seems to be present in most of species in order Solanales. My suggestion is that the authors should enrich their work with biochemical data on more key enzymatic steps based on their bioinformatic analysis.

Reply: Thank you for your valuable suggestions. Utilizing the standard calibration production of the *TRI* gene, we conducted gain-of-function experiments on the *TRI* gene from a non-HS-producing potato, as well as loss-of-function experiments of the HS-producing species involving forward and reverse mutations at multiple sites. The experimental results confirmed our predictions based on bioinformatic analyses. Additionally, we investigated the functionality of another crucial HS gene, the *LS* gene, from two HS-producing species, *A. tanguticus* and *B. arborea*. As anticipated, the *LS* genes from these two species exhibited high activity, aligning with our predictions from bioinformatic analyses (please refer to the main context in the revised manuscript). Furthermore, we performed functional experiments on the *PPAR*, *CYP80F1*, *HDH*, and *H6H* genes (see Figs. 5 and 6) and validated the roles of *TRI*, *LS*, *HDH*, and *H6H* genes in HS biosynthesis in plants using VIGS technology (Supplementary Figure 18). The detailed procedures and results can be found in the revised manuscript.

These genes primarily reside downstream of the HS pathway and maintain collinearity with only a few non-HS-producing species (see Fig. 4). Our bioinformatic analyses and functional experimental results are in perfect agreement, providing compelling evidence that the HS-synthesis pathways most likely originated in the common ancestor of all Solanaceae lineages. Furthermore, the loss of genes in non-HS-producing tribes has contributed to the limited distribution of HS in the Solanaceae family.

Reviewer #2 (Remarks to the Author):

The manuscript entitled “Multiple biosynthesis losses of two specific tropane alkaloids in the Solanaceae” tries to shed light on the biosynthesis of tropane alkaloids specifically hyoscyamine and scopolamine. To this aim, the authors sample within the Solanaceae and sequence three genomes of species that produce these tropane alkaloids (*Anisodus tanguticus*, *Brugmansia arborea*, *Mandragora caulescens*) and one (*Lycium chinense*) that does not. They then compare the genomes to existing genome assemblies of Solanaceae species and determine that the pathways generally seem to be there. Indeed, cloning a potato TRI homolog (i.e. a non-producer) did not show activity. In total the authors produce some interesting evidence but there might be issues with the genome analysis see below.

The main interesting findings were

- 1) chromosome level assemblies of all four species
- 2) An identification of likely whole genome duplication events
- 3) Demonstration that no activity could be found for potato TRI and identification of two crucial sites for TRI synthesis
- 4) Demonstration that the tropane pathway was likely ancestral and was subsequently lost in some species.

Comments

- 1) One finding are the WGD events. Whilst the number of chromosomes make these very likely, there are some issues which should be checked not only for the sake of the WGD analysis, but to obtain a decent genome annotation and to investigate these genomes in the future as a resource.

Here are the observations:

1. *L. chinense* is the most heterozygous genome with ca 0.9% het estimated.

Reply: The level of genomic heterozygosity may vary greatly among different plant species. In plants, the average level of genomic heterozygosity varies between 1 and 2%. Here are a few examples to illustrate that eggplant genome have heterozygosity of 0.067%¹⁷, Pepino genome of 0.83%¹⁸ and the 12~40.5% of the different populations of *Cannabis sativa*¹⁹.

2. root mapping rates for RNAseq data in Brugmanisa is low S table 8

Reply: We have recalculated the mapping rate of transcriptome data on four genomes using HISAT software. The new statistical results are provided in Supplementary Table 8 and highlighted in yellow.

3. BUSCO and some annotation statistics. Species duplicated BUSCO S Table 10 Total genes RVM S Table 11ff Annotated NR S Table 17 *Anisodus tanguticus* 28.5 % 46606 92.1% *Brugmansia arborea* 3.9% 32347 92.2% *Lycium chinense* 12.3% 54946 79% *Mandragora caulescens* 14.1% 29193 93.8%. Now *Mandragora* and *Anisodus* should have seen a WGD however the numbers for BUSCO duplications are relatively low, but this could be due to an older WGD event which can be discussed and should be looked at. However, the high value of 12.3 % duplicated BUSCO in *Lycium* cannot be explained, unless these are unremoved haplotypes (see het rate above) as *Lycium* is supposed not to have a WGD. This is somewhat supported by gene number as well (too high), but given that *Lycium* has a much lower NR similarity rate (same for Swissport) in addition there seem to be an overannotation of genes potentially.

Reply: Haplotype divergence in regions of high heterozygosity may lead to the assemblies of two copies instead of one copy. We used the hifiasm software to assemble the HiFi reads of *L. chinense* and *M. caulescens*. This software automatically eliminates duplicated haplotypes. Since the *L. chinense* genome exhibited the highest heterozygosity among the four assemblies that we sequenced, we performed additional haplotype-removal process using `purge_dups`²⁰. Despite this, only 3.5 Mbp of haplotig sequences were removed, compared to a total genome size of 1.5 Gbp. This indicates a low level of haplotype duplications in the *L. chinense* assembly. Our focus was on the duplicated BUSCO orthologs in *L. chinense* and *B. arborea*, which accounted for 13.1% (211) and 4.3% (70) of the total respectively (the BUSCO protein model, Supplementary Table 16). Among these duplicates, 160 in *L. chinense* and 52 in *B. arborea* were derived from the Solanaceous WGT events, respectively, accounting for 9.91% and 3.22% of their duplicated BUSCO orthologs. *L. chinense* exhibited a higher proportion of BUSCO orthologs due to the whole WGT events and a high value of duplicated BUSCO. Although the NR and Swiss-Prot annotations were relatively lower compared to the other three species, nearly 92% of the total predicted genes were assigned to entries in seven functional databases (Supplementary Table 17).

4. The low mapping rates for Brugmansia data however are not reflected in the BUSCO or annotation rates, here one could only speculate for an issue in RNAseq.

Reply: We have recalculated the mapping rate of transcriptome data on *Brugmansia arborea* genome assembly. The new statistical results are provided in Supplementary Table 8 and highlighted in yellow.

5. In general as also shown in S Table 15, at the very least the gene numbers for *L. chinense* are too high. For *A. tang.* And *M. caul.* (WGD and 24 chromosomes!) it would make sense to investigate why the gene numbers are not even close to 2x that of potato and tomato and in the case of *M. caul.* even less than 1x potentially indicating an underannotation. General gene structural annotation is not too easy and many ab-initio tools overannotate. But functional sub-annotation or comparison with references can also help improve this.

Reply: *M. caulescens* and *L. chinense* belong to the Mandragoreae and Lycieae, respectively, and potato and tomato are the members of the Solaninae. They are distantly related; thus, it is not appropriate to compare gene numbers between them. In fact, the occurrence of the polyploidy in plant evolution will lead to dramatic chromosomal breaks, fusions and loss of chromosomal fragments, which may lead to the large differences in genome size (*L. chinense*-1.5Gbp, *M. caulescens*-711Mbp) and the number of genes between species.

6. Data must be made available

Reply: When the manuscript is accepted, all data will be released immediately.

7. As the authors identified potential potato TRI residues that seem to be important for activity would it not make sense to consider the activity of the potato TRI mutant A109V and L167V as well, i.e. Creating a gain of function mutant? Even though the loss of function study is already interesting. (But we know they are necessary for function but are these two residues sufficient as well?)

Reply: Thank for your valuable suggestions. It is well worth doing as you suggested. We spent several months (more than six months by three postdocs and PhD students) to do this gain-of-function mutant. The results confirmed our prediction from bioinformatic analyses. We mutated four key sites of the non-HS-producing potato *TRI* gene *SrTRI*, Ala¹⁰⁹, Val¹⁵⁵, Leu¹⁶⁷ and Ala²⁰¹, into Val¹⁰⁹, Leu¹⁵⁵, Val¹⁶⁷ and Gly²⁰¹ of the HS-producing *TRI* gene, *BaTRI*. Then, we analyzed the

enzymatic kinetics of *SrTRI*^{A109V, L167V} and *SrTRI*^{A109V, V155L, L167V, A201G}. As expected, the *Kcat/Km* value of *SrTRI*^{A109V, L167V} and *SrTRI*^{A109V, V155L, L167V, A201G} increased markedly compared with *SrTRI*, followed by the gained ability to catalyze tropinone to tropine (Figs. 5c-5e, Supplementary Tables 27, Supplementary Figure 33).

Minor and typos

8. Page 4 line 100: “*Lycium chinense*, 102 respectively (Table 1, Supplementary Fig. 1,” that is S Fig 2

Reply: Done.

9. Page 5 line 133: in the three gene sets: should likely be in the four gene sets

Reply: Done.

10. Page 14 line 381 Pacbio raw reads = probably PacBio HIFI reads are meant (i.e. preprocessed data)

Reply: Done.

11. Page 15 line 390 HiC data is given as approx. 1Gb, 525Mb 377MB and 273MB I think the numbers are reads numbers (HiC data in this volume would not be enough to scaffold genomes)

Reply: Done.

12. Page 18 line 476 “Transcripts were assembled and quantified using Stringtie” Even though Stringtie produces high quality annotations, I would still use the final genome annotation results and not interim Stringtie results. But take the potential issues from above.

Reply: Yes, we used the final genome annotation results. After filtering low quality reads, the quality checked RNA-seq reads from different tissues were aligned to four assembled genomes using HISAT2. The transcripts per kilobase of exon model per million mapped reads (TPM values) was calculate performed by StringTie. To facilitate subsequent analysis, here we only selected the finally annotated genes for calculation using the parameter ‘-e’, and did not assemble new transcripts. We added all descriptions.

13. Unfortunately checking is not possible as data is not available.

Reply: When the manuscript is accepted, all data will be released immediately.

14. *Brugmansia* was shown (? Or assembled) to 13 chromosomes. Is there evidence for the deviation from the base numbers. There seems to be little cytological data but the data available seems to indicate *Brugmansia arborea* (previously *D. arborea*) had n=12 chromosomes see <https://doi.org/10.1080/00087114.1968.10796312>. That said it could be earlier species misidentifications from the *Datura* genus as well but needs to be discussed + checked at least.

Reply: A study on karyotype analysis of the Solanaceae species showed that the karyotype of *Brugmansia arborea* was 2n=26, so it is finally attached with 13 pseudo-chromosomes²¹. (<http://hdl.handle.net/10603/48057>). We added this information in the revised manuscript.

15. S figure 9 : Boxes seem to be drawn arbitrarily, compare a and b. First box in b is also visible in a. This should be done more exhaustively

Reply: We delete these boxes.

16. Throughout the manuscript leaded = led

Reply: Revised.

Reviewer #3 (Remarks to the Author):

The present manuscript describes a part of the evolutionary history of tropane alkaloid (TA) biosynthesis in the Solanaceae family. While focusing on hyoscyamine and scopolamine synthesis,

the authors sequenced the genome of three species producing these two TAs and one non-producing species. By combining this new dataset to 11 previously sequenced genomes, authors analyzed the synteny of the hyoscyamine and scopolamine biosynthetic pathway and established the high collinearity between the corresponding genes. This allowed them to propose an ancestral origin of this TA biosynthetic pathway before lineage diversification, accompanied by multiple losses (eg: exon losses) leading to pseudogenization and loss of hyoscyamine and scopolamine biosynthesis capacity. Such a notion was also reinforced by the characterization of the mutations in the sequence of the Tropinone reductase I that did not let to activity loss in some hyoscyamine and scopolamine nonproducing plant species. In sum, this is an informative describing work. In addition, experiments have been well conducted, the results are convincing, and the manuscript is well written. Comments and Concerns are listed below:

1. Unify the term specialized and secondary metabolites

Reply: Changed.

2. Both DNA and RNA extraction protocols should be described. HiC library preparation should be detailed as well.

Reply: Thank you for your suggestions and we have added these descriptions. It is detailed in line 454-456, line 471-473 and line 492-500 of the revised manuscript.

3. Long-read library preparation should be described prior giving sequencing results

Reply: Thank you for your suggestions and we have made adjustment. It is detailed in line 461-466 of the revised manuscript.

4. Cufflinks is considered depreciated now as there is no maintenance for 10 years. other tools should be preferred (StringTie, Trinity...)

Reply: In the transcriptome-based prediction of the *A. tanguticus* gene structure, we used two methods. One was by mapping the RNA-seq data to the genome and assembling the transcripts using Tophat and Cufflinks. The other was by applying Trinity to *de novo* assemble the RNA-seq data followed by PASA to improve the gene structures. Considering that Cufflinks is depreciated, we applied the Trinity to assemble the RNA-seq data for other three genomes.

5. Was trinity run on its own or using the genome-guided option?

Reply: Trinity provides two ways to use it, one is *de novo* assemble without reference genome, and the other is mapping RNA-seq onto genome then assemble with genome-guided. In our manuscript, we used Trinity based on *de novo* assemble option.

6. Please describe RNA extraction method

Reply: Thank you for your suggestions and we have added this description. It is detailed in line 471-473 of the revised manuscript.

7. Please precise if StringTie assembly was performed individually and merged (using stringtie -merge) or using all RNAseq data together.

Reply: After filtering low quality reads, the quality checked RNA-seq reads from different tissues were then aligned to the four assembled reference genomes using HISAT2. The Binary Alignment Map (BAM) results were used as input for StringTie with default parameters for genome-based transcript assembly. The transcript from each tissue are merged and filtered using a string -merge module, resulting in a merged transcript. The merged transcript was then used as a reference for reassemble the transcripts and calculate abundance for each tissue.

8. Be consistent in the tools you used: was genome-guided transcriptome assembly performed using cufflinks (L424) or StringTie (L476)?

Reply: In transcriptome-based prediction of gene structure, Tophat and Cufflinks software were used for genome-based transcript assembly, followed by PASA to improve the gene structures. StringTie was used to calculate transcripts per kilobase of exon model per million mapped reads (TPM values) for each sample. It is detailed in line 533-536 and 587-590 of the revised manuscript.

9. LL114-117: four genomes are listed but only three %

Reply: Added.

10. LL359-361 “We obtained polymerase reads totaling ~164Gb, ~450Gb, ~477Gb and ~87Gb from PromethION flow cells on Nanopore and SMRT cells on PacBio Sequel II for *A. tanguticus*, *L. chinense*, *B. arborea* and *M. caulescens* respectively.” You should rephrase as it is unclear which species has been sequenced with ONT and PacBio.

Reply: Thanks for your reminder. We modified our statements. We obtained polymerase reads totaling ~164Gb from PromethION flow cells on Nanopore for *A. tanguticus*, and ~450Gb, ~477Gb and ~87Gb from SMRT cells on PacBio Sequel II for *L. chinense*, *B. arborea* and *M. caulescens*, respectively. It is detailed in line 466-469 of the revised manuscript.

11. L364: please precise NovaSeq 6000

Reply: Done.

12. L371: please be consistent when precisising the PacBio Sequel you used.

Reply: Done.

13. All versions of software and parameters should be mentioned.

Reply: Thank you for your suggestions and we have added it.

14. The authors have mutated two key residues of BaTRI leading to a dramatic loss of activity. Did they consider performing the reverse mutations in StTRI? This would be helpful to determine the role of this mutation in the loss of activity. Indeed, StTRI and LcTRI bear the same residues but different activities.

Reply: Thank for your valuable suggestions. We spent several months to do this gain-of-function mutant and functional tests. The results confirmed our prediction from bioinformatic analyses. We mutated four key sites of the non-HS-producing potato *TRI* gene *StTRI*, Ala¹⁰⁹, Val¹⁵⁵, Leu¹⁶⁷ and Ala²⁰¹, into Val¹⁰⁹, Leu¹⁵⁵, Val¹⁶⁷ and Gly²⁰¹ of the HS-producing *TRI* gene, *BaTRI*. Then, we analyzed the enzymatic kinetics of *StTRI*^{A109V, L167V} and *StTRI*^{A109V, V155L, L167V, A201G}. As expected, the *Kcat/Km* value of *StTRI*^{A109V, L167V} and *StTRI*^{A109V, V155L, L167V, A201G} increased markedly compared with *StTRI*, followed by the gained ability to catalyze tropinone to tropine (Figs. 5c-5e, Supplementary Tables 27, Supplementary Figure 33).

Related References:

1. Hashimoto, T., Nakajima, K., Ongena, G., Yamada, Y. Two tropinone reductases with distinct stereospecificities from cultured roots of *Hyoscyamus niger*. *Plant Physiol.* **100**, 836-845 (1992).
2. Rajewski, A., Carter-House, D., Stajich, J. & Litt, A. *Datura genome* reveals duplications of psychoactive alkaloid biosynthetic genes and high mutation rate following tissue culture. *Bmc Genomics* **22**, (2021).
3. Zhang, F. *et al.* Revealing evolution of tropane alkaloid biosynthesis by analyzing two genomes in the Solanaceae family. *Nat. Commun.* **14**, (2023).
4. Drost-Karbowska, K., Hajdrych-Szauffer, M. & Kowalewski, Z. Search for alkaloid-type bases in *Lycium halimifolium*. *Acta poloniae pharmaceutica* **41**, 127-129 (1984).
5. Harsh, M. L. Tropane Alkaloids from *Lycium barbarum* Linn In vivo and In vitro. *Curr. Sci. India.* **58**, 817-818 (1989).

6. Lu, A. M. & Wang, M. L. On the identification of the original plants in the modernization of Chinese herbal medicine-An example from the taxonomy and exploitation of Gouqi. *Acta. Bot. Boreal. Occident. Sin.* **43**, 21-27 (2003).
7. White, A. Towards a safer choice: the practice of traditional Chinese medicine in Australia. *University of Western Sydney Macarthur* **2**, 118 (1996).
8. Adams, M., Wiedenmann, M., Tittel, G. & Bauer, R. HPLC-MS trace analysis of atropine in *Lycium barbarum* berries. *Phytochem Analysis* **17**, 279-283 (2006).
9. Yao, X. *et al.* HPLC-MS trace analysis of atropine in different *Lycii fructus* samples. *Lishzhen medicine and materia medica reserch. China.* **22**, 2 (2011).
10. Kokotkiewicz, A. *et al.* Densitometric TLC analysis for the control of tropane and steroidal alkaloids in *Lycium barbarum*. *Food Chem.* **221**, 535-540 (2017).
11. Zhao, W. H. & Shi, Y. P. Simultaneous quantification of three tropane alkaloids in goji berries by cleanup of the graphene/hexagonal boron nitride hybrids and ultra-high-performance liquid chromatography tandem mass spectrometry. *J. Sep. Sci.* **43**, 3636-3645 (2020).
12. Wu, Y. B. *et al.* Antiosteoporotic Activity of Anthraquinones from *Morinda officinalis* on Osteoblasts and Osteoclasts. *Molecules* **14**, 573-583 (2009).
13. Zeng, L. *et al.* Alternative pathway to the formation of transcinnamic acid derived from L-phenylalanine in tea (*Camellia sinensis*) plants and other plants. *J. Agric. Food Chem.* **68**, 3415–3424 (2020).
14. Liu, T., Zhu, P., Cheng, K. D., Meng, C., & He, H. X. Molecular cloning, expression and characterization of hyoscyamine 6beta-hydroxylase from hairy roots of *Anisodus tanguticus*. *Planta Med.* **71**, 249-253 (2005).
15. Qiang, W., Hou, Y. L., Li, X., K, X., & Liao, Z. H. Cloning and expression of the key enzyme hyoscyamine 6 beta-hydroxylase gene (*DaH6H*) in scopolamine biosynthesis of *Datura arborea*. *Acta Pharm Sin B.* **50**, 1346 (2015).
16. Qiu, F. *et al.* Biochemical and metabolic insights into hyoscyamine dehydrogenase. *Acs. Catal.* **11**, 2912-2924 (2021).
17. Barchi, L. *et al.* A chromosome-anchored eggplant genome sequence reveals key events in Solanaceae evolution. *Sci. Rep.* **9**, 11769 (2019).
18. Song, X. *et al.* Chromosome-level pepino genome provides insights into genome evolution and anthocyanin biosynthesis in Solanaceae. *Plant J.* **110** (2022).
19. Hurgobin, B. *et al.* Recent advances in Cannabis sativa genomics research. *New Phytol.* **230**, 73-89 (2021).
20. Guan, D. *et al.* Identifying and removing haplotypic duplication in primary genome assemblies. *Bioinformatics* **36**, 2896-2898 (2020).
21. Jagatheeswari, D. Cytotaxonomical and tissue culture studies on Some members of Solanaceae. (2014).

Reviewers' Comments:

Reviewer #1:

Remarks to the Author:

The current manuscript is a resubmission after 9 months and during the first review I have expressed my concerns that the efforts by the authors to explain the evolution of tropane metabolism in family Solanaceae due to the absence of biochemical data, since there was an assumption that syntenic relationships are connected to conserved activity without including some biochemistry data.

I have to congratulate the authors that they added in the resubmission a substantial number of enzyme assays.

I have asked to improve the introduction to tropane metabolism and the introductory figure. Authors did not improve the narrative of introduction and the new figure is of lower quality. In general, I found all the figures of low quality.

I really appreciated that they used the two genomes of *Ipomoea* species from the family Convolvulaceae as outgroup as I have implied.

They address my question and the confusion caused in literature regarding the presence of tropane alkaloids in *Lycium* sp. However the figures FigR3, FigR4, FigR5 are of low quality.

Since the submission of the first draft last December, two major papers have been published, regarding evolution of tropane alkaloids. One addressing the convergence between the tropane biosynthetic pathway in Solanaceae and Erythroxylaceae

<https://www.pnas.org/doi/10.1073/pnas.2302448120>. The second paper (Zhang, F. et al. Revealing evolution of tropane alkaloid biosynthesis by analyzing two genomes in the Solanaceae family. *Nat. Commun.* 14, (2023)) is published in Nature Communications and it was submitted probably the same time as the current manuscript. The paper by Zhang et al published in Nature Communications and the current manuscript examined the same question about the evolution of tropane alkaloids biosynthesis in different lineages in Solanaceae by sequencing selectively specific plant species.

After the publication of the two above mentioned papers, I was literally expecting the authors to adjust their story more relevant to the current status of knowledge.

The current manuscript reports the following experimental data:

1. The genomes at chromosome level of three species producing the medicinally important tropane alkaloids hyoscyamine and scopolamine from family Solanaceae: *Anisodus tangiticus*, *Brugmansia arborea*, and *Mandragora cualescens*. Additionally, it is the genome of *Lycium chinense*, a plant from family Solanaceae which does not produce either hyoscyamine and scopolamine. The four species are representing four different tribes in Solanaceae.
2. Standard/typical genomic analysis and comparison (mainly syteny).
3. Classical enzyme assays (with protein purification) of five TRI enzymes from five plants representing different five different subfamilies in Solanaceae.
4. Enzyme activity of two LS enzymes only from *Anisodus tangiticus* and *Brugmansia arborea* by transient expression in host plant *N. benthamiana* (why not *M. cualescens*?)
5. Classical enzyme assays (with protein purification) of PPAR enzymes however it is unclear which enzymes from which species were used (at line 670 authors describe as "respective species").
6. Combined enzyme activity of CYP80F1 and HDH from *A. tangiticus*, *B. arborea*, and *M. cualescens* by transient expression in host plant *N. benthamiana*.
7. In vitro enzyme activity of H6H enzymes from *L. chinense* and *M. cualescens*.
8. VIGS of TRI, LS, HDH and H6H genes from *A. tangiticus*.
9. LCMS analysis and detection of tropine, littorine, phenyl-lactate and hyoscyamine from (? *Lycium chinense* – it is not well stated, and the figures are rather not at high standard).
10. Inference of ancestral states (?) of key biosynthetic genes. Though I don't see anywhere in manuscript any inference of ancestral state. All these data reflect a substantial amount of work.

Overall, by reading the text it is not clear to me what is the major biological question, what is the major hypothesis behind this work. In certain extend, the lack of a clear major question/ hypothesis is enhanced by the manuscript structure, convoluted narrative and not clear presentation of experimental results. The whole reading experience is getting worse with the rather bad quality of

figures and poor presentation of experimental data. Additionally, the poor quality of English does not help.

I must point out that the presented comparative genomic analysis is mainly inside family Solanaceae and it cannot be at any point be considered as comparative genomic analysis among distant species. In the introduction of paper there is some conversation about convergent functions and independent origins possibly inspired by the convergence of metabolic pathways of tropane alkaloids in two different and distant plant families, Erythroxylaceae and Solanaceae. While the alternative which is proposed is the loss of gene or of enzymatic function. There are some recently published papers eg Li et al Mol Plant 2023 which show that the absence of specific metabolites between closely related species has nothing to do with loss of enzymatic activity, pseudogenization or gene loss and it might be more complicated. There are some recent papers (some of them are cited) which use phylogenomics combined with biochemistry to discover interesting aspects in evolution of pathways in specialised metabolism.

I am not sure what means the expression "forward and reverse mutations" but I found the conclusion of mutational work is rather trivial. If the presence or not of biosynthetic pathway of hyoscyamine and scopolamine in specific lineages at tribe/genera/species in Solanaceae is result of enzyme loss of function, pseudogenization or gene loss is probably one of the most expected (and less exciting) explanations.

There are plenty of works (including papers published in the current journal the last 4 years) showing the role of phenomena like enzyme loss of function, pseudogenization or gene loss in absence of specific of specialized metabolites.

Overall, a lot of work which is done relatively well and address my technical concerns during the first round of review. But there is no biological question or hypothesis and the conclusions drawn are neither exciting or unexpected.

Independently to the other two papers published about the evolution of tropane metabolism, I do not see any novelty in the current paper.

Reviewer #2:

Remarks to the Author:

The authors have done a tremendous job of analyzing the biology of the TRI gene and also performed gain of function experiments in potato.

Secondly they added measures on wolfberry.

The manuscript has greatly increased in value.

My comment regarding the number of duplications was a cautionary one, in our hand also purging duplicates does not always get rid of all resolved haplotypes leading to duplications. But it is a step in the right direction. A simple solution would be to point out in the discussion that there might be remaning issues.

Reviewer #3:

Remarks to the Author:

In the revised version of their manuscript describing the evolutionary history of tropane alkaloid biosynthesis in the Solanaceae family, the authors have addressed all my concerns. I have especially appreciated their efforts to generate the gain-of -function mutant and its characterization. The whole story is interesting and the numerous corrections they did considerably improve this version of the manuscript.

Response to the Reviewer 's Comments:

Reviewer #1 (Remarks to the Author):

The current manuscript is a resubmission after 9 months and during the first review I have expressed my concerns that the efforts by the authors to explain the evolution of tropane metabolism in family Solanaceae due to the absence of biochemical data, since there was an assumption that syntenic relationships are connected to conserved activity without including some biochemistry data.

I have to congratulate the authors that they added in the resubmission a substantial number of enzyme assays.

Reply: Thank you so much for appreciating our tremendous efforts and work, which were carried out under your guidance and valuable suggestions. Over the course of the past nine months, two postdoctoral researchers and two PhD students have tirelessly worked day and night to establish the molecular analyses of HS in the alpine plant *Anisodus tanguticus*, a task that proved to be extremely challenging. Through the development of this new molecular system, we have successfully addressed all the significant questions that had been raised previously. We are extremely grateful for your recognition of our dedicated efforts.

I have asked to improve the introduction to tropane metabolism and the introductory figure.

Reply: We have now revised the introduction of the metabolic pathway of tropanes alkaloids and optimized the pathway of HS, that was as shown in Supplementary Figure 1. In addition, regarding your suggestion that TRI should be included as a part of tropane ring biosynthesis in Figure 4, we have also made corresponding modifications.

Authors did not improve the narrative of introduction and the new figure is of lower quality. In general, I found all the figures of low quality.

Reply: We sincerely appreciate your valuable comments, as they have provided us with highly relevant and helpful suggestions for our paper. As a result, we have taken great care to revise the introduction and make corresponding modifications in order to enhance both its writing and logical coherence. Furthermore, we have diligently reviewed and revised all the figures throughout the manuscript (including supplementary Figure), with the goal of ensuring accuracy, conciseness, and visual clarity in their presentations. In our previous submission, we followed the journal's guidelines by submitting low-quality figures for easier uploading and opening. Therefore, in this revised version, we have included relatively large-size figures to maintain clearer visuals.

I really appreciated that they used the two genomes of Ipomoea species from the family Convolvulaceae as outgroup as I have implied.

Reply: We sincerely appreciate your valuable suggestions and guidance.

They address my question and the confusion caused in literature regarding the presence of tropane alkaloids in Lycium sp. However, the figures FigR3, FigR4, FigR5 are of low quality.

Reply: Thank you so much for appreciating our tremendous efforts and work to solve this confusion. We have revised figures FigR3, FigR4, FigR5, also known as attached Supplementary Figure 36.

Since the submission of the first draft last December, two major papers have been published, regarding evolution of tropane alkaloids. One addressing the convergence between the tropane

biosynthetic pathway in Solanaceae and Erythroxylaceae <https://www.pnas.org/doi/10.1073/pnas.2302448120>. The second paper (Zhang, F. et al. Revealing evolution of tropane alkaloid biosynthesis by analyzing two genomes in the Solanaceae family. *Nat. Commun.* 14, (2023)) is published in *Nature Communications* and it was submitted probably the same time as the current manuscript. The paper by Zhang et al published in *Nature Communications* and the current manuscript examined the same question about the evolution of tropane alkaloids biosynthesis in different lineages in Solanaceae by sequencing selectively specific plant species. After the publication of the two above mentioned papers, I was literally expecting the authors to adjust their story more relevant to the current status of knowledge.

Reply: Thank you so much for these suggestions. We have now revised our manuscript to place greater emphasis on genomic comparison, biochemical evidence, and the potential for future HS production through genetic modifications. Prior to submitting our manuscript to *Nature Communications*, we consulted with the editor and carefully reviewed the submission guidelines. It is important to note that all unpublished papers with similar results submitted to *Nature*-serial journals can be considered as back-to-back submissions if similar results are published after each initial submission. The priorities of each respective submission are guaranteed. Here, we have provided two pairs of back-to-back research studies published in the *Nature*-serial journals by different research groups at different times. There are many such examples. Therefore, our results and those two papers published in *PNAS* and *Nature Communications* can be considered as back-to-back submissions to confirm results of each another with the similar situation.

- 1) The slow-evolving *Acorus tatarinowii* genome sheds light on ancestral monocot evolution. **July 14, 2022. Nature Plant** vs Diploid and tetraploid genomes of *Acorus* and the evolution of monocots. **June 2023, Nature communications**
- 2) *Gossypium barbadense* and *Gossypium hirsutum* genomes provide insights into the origin and evolution of allotetraploid cotton. **March 18, 2019 Nature genetics** vs Genome sequence of *Gossypium herbaceum* and genome updates of *Gossypium arboreum* and *Gossypium hirsutum* provide insights into cotton A-genome evolution. **April 13, 2020 Nature genetics**

The current manuscript reports the following experimental data:

1. The genomes at chromosome level of three species producing the medicinally important tropane alkaloids hyoscyamine and scopolamine from family Solanaceae: *Anisodus tangiticus*, *Brugmansia arborea*, and *Mandragora culaesens*. Additionally, it is the genome of *Lycium chinense*, a plant from family Solanaceae which does not produce either hyoscyamine and scopolamine. The four species are representing four different tribes in Solanaceae.
2. Standard/typical genomic analysis and comparison (mainly synteny).
3. Classical enzyme assays (with protein purification) of five TRI enzymes from five plants representing different five different subfamilies in Solanaceae.
4. Enzyme activity of two LS enzymes only from *Anisodus tangiticus* and *Brugmansia arborea* by transient expression in host plant *N. benthamiana* (why not *M. culaesens*?)

Reply: Due to the challenges associated with obtaining samples of the alpine *M. culaesens*, we were unable to conduct LS enzyme activity experiments simultaneously for it and *Anisodus tangiticus* and *Brugmansia arborea*. When we obtained the purified LS protein from *M. culaesens* and prepared for the enzyme activity assay, we discovered that the required littorine standard for the

experiment had been depleted. Unfortunately, it was not feasible to procure a standard that met the necessary purity criteria within a reasonable timeframe. We considered exploring alternative methods such as chemical synthesis; however, the estimated time required for this approach was highly uncertain. Additionally, extensive bioinformatics analyses revealed that the *LS* genes of *Anisodus tangiticus*, *Brugmansia arborea*, and *M. cualesens* share identical structural domains and highly conserved sequences. Based on this finding, we inferred that all three enzymes should possess the same catalytic ability to synthesize littorine from the substrate. Consequently, in this study, we have exclusively presented the LS enzyme activity results for *Anisodus tangiticus* and *Brugmansia arborea* only. We believe that the absence of LS enzyme activity data for *M. cualesens* will not impact our conclusions as a bioinformatic manuscript. In fact, similar back-to-back papers (e.g., Zhang et al., 2023, Nature Communications) have also not examined all enzyme activities for all species discussed.

5. *Classical enzyme assays (with protein purification) of PPAR enzymes however it is unclear which enzymes from which species were used (at line 670 authors describe as “respective species”).*

Reply: Thank you for your valuable advice. We have clarified these issues. In line 338 of the manuscript, the description of PPAR enzyme experiment results explicitly mentions that we conducted enzyme activity analysis on PPAR enzymes of *B. arborea*, *M. caulescens*, and *P. inflata* species, which showed their ability to catalyze phenylpyruvic acid into phenyllactate. Initially, our intention was to perform enzyme activity experiments using collinear PPAR genes from all species analyzed. However, similar to previous studies on PPAR enzymes, the purification of PPAR enzymes proved to be extremely challenging. Despite multiple attempts, we were only able to obtain purified PPAR proteins from *B. arborea*, *M. caulescens*, and *P. inflata*. Therefore, the experimental results presented in the paper are limited to these three enzymes.

6. *Combined enzyme activity of CYP80F1 and HDH from A. tangiticus, B. arborea, and M. caulescens by transient expression in host plant N. benthamiana.*

7. *In vitro enzyme activity of H6H enzymes from L. chinense and M. caulescens.*

8. *VIGS of TRI, LS, HDH and H6H genes from A. tangiticus.*

9. *LCMS analysis and detection of tropine, littorine, phenyl-lactate and hyoscyamine from (? Lycium chinense – it is not well stated, and the figures are rather not at high standard).*

Reply: Thank you for bringing this to our attention. In lines 789-794 of the manuscript, we stated the HPLC-MS analysis of *Lycium chinense*. Furthermore, addressing the issue of inadequate presentation of the figures in that section, we have made the necessary revisions. Please refer to Supplementary Figure 36 for the revised manuscript.

10. *Inference of ancestral states (?) of key biosynthetic genes. Though I don't see anywhere in manuscript any inference of ancestral state. All these data reflect a substantial amount of work.*

Reply: Thank you for raising this concern. We have addressed the issue and made revision to improve the clarity of the wording in lines 796-800 of the manuscript, where we described the method used for the ancestral sequence reconstruction of the *TRI* gene. Additionally, in lines 282-284 of the manuscript, we have provided a description of the results obtained from simulating ancestral sites of the *TRI* gene.

Overall, by reading the text it is not clear to me what is the major biological question, what is the major hypothesis behind this work. In certain extend, the lack of a clear major question/ hypothesis is enhanced by the manuscript structure, convoluted narrative and not clear presentation of experimental results. The whole reading experience is getting worse with the rather bad quality of figures and poor presentation of experimental data. Additionally, the poor quality of English does not help. I must point out that the presented comparative genomic analysis is mainly inside family Solanaceae and it cannot be at any point be considered as comparative genomic analysis among distant species.

Reply: Thank you for expressing your concerns. We have revised the overall structure and argumentation of the manuscript. In the introduction, we clearly stated our aim to discern between two hypotheses: convergent evolution or loss of HS biosynthesis gene following an ancestral origin. This question is of significant biological importance and is applicable to various traits, including specific secondary metabolites. We have incorporated numerous functional tests of the examined genes as per your requirements. However, due to the extensive addition of these experimental results, the manuscript may be slightly longer compared to other similar bioinformatic papers. Nonetheless, we believe that these comprehensive experiments are necessary to strengthen all our conclusions as you suggested before. We have simplified the discussion of experimental results to improve clarity, and we have also enhanced the quality of all figures (including supplementary Figure). The entire manuscript has been proofread by an American professor with a native language in this field to ensure proper English usage.

Regarding the issue you raised about the comparative genomics analysis between Solanaceae species not considering them as distant species comparative genomes, we would like to provide the following explanation:

Our study primarily focuses on the evolutionary mechanisms underlying the restricted distribution of two specific tropane alkaloids, hyoscyamine and scopolamine, within the Solanaceae family. Therefore, the species selected for analysis are limited to the Solanaceae family, with the exception of the outgroup from the Convolvulaceae family. The three tribes studied, namely Hyoscyaminae (*Anisodus tanguticus*), Datureae (*Brugmansia arborea*), and Mandragorinae (*Mandragora caulescens*), are widely recognized as distant lineages within the phylogeny of Solanaceae. Furthermore, it is important to note that our article consistently emphasizes the focus on studying species within the Solanaceae family. Therefore, the term "distant species" used here is a relative description rather than an absolute one. Here the species within the same tribe can be treated as the closely related species while those in different tribes should be called as the distant species. Similar statements are also supported in a back-to-back study (Zhang et al. 2023) through the analysis of two species from the same distant lineage.

In the introduction of paper there is some conversation about convergent functions and independent origins possibly inspired by the convergence of metabolic pathways of tropane alkaloids in two different and distant plant families, Erythroxylaceae and Solanaceae. While the alternative which is proposed is the loss of gene or of enzymatic function. There are some recently published papers eg Li et al Mol Plant 2023 which show that the absence of specific metabolites between closely related species has nothing to do with loss of enzymatic activity, pseudogenization or gene loss and it might be more complicated. There are some recent papers (some of them are cited) which use phylogenomics combined with biochemistry to discover interesting aspects in evolution of pathways

in specialised metabolism.

Reply: Yes, you are correct that the absence of specific metabolites between closely related species cannot be solely attributed to loss of enzymatic activity, pseudogenization, or gene loss. The underlying factors influencing this phenomenon are likely more complex. We have revised our related statements to reflect this understanding.

I am not sure what means the expression “forward and reverse mutations” but I found the conclusion of mutational work is rather trivial. If the presence or not of biosynthetic pathway of hyoscyamine and scopolamine in specific lineages at tribe/genera/species in Solanaceae is result of enzyme loss of function, pseudogenization or gene loss is probably one of the most expected (and less exciting) explanations.

Reply: Thank you for bringing this to our attention. Indeed, our choice of words was not sufficiently professional and clear. We have revised "forward and reverse mutations" to "gain-of-function and loss-of-function mutations" accordingly.

There are plenty of works (including papers published in the current journal the last 4 years) showing the role of phenomena like enzyme loss of function, pseudogenization or gene loss in absence of specific of specialized metabolites.

Overall, a lot of work which is done relatively well and address my technical concerns during the first round of review. But there is no biological question or hypothesis and the conclusions drawn are neither exciting or unexpected. Independently to the other two papers published about the evolution of tropane metabolism, I do not see any novelty in the current paper.

Reply: We greatly appreciate the recognition of our efforts in addressing your previous concerns. As we have previously mentioned, we have cited relevant papers that discuss the absence of specific metabolites resulting from enzyme loss of function, pseudogenization, or gene loss. Each of these papers may not present groundbreaking findings on its own. However, they together reveal evolutionary diversity and complexity of the origins of the secondary metabolites. Additionally, as we stated before, Nature Communications upholds the priorities of each submission even if similar results are published subsequently. Therefore, our work can be considered as a series of back-to-back studies in conjunction with two recent papers published in PNAS and Nature Communications. These back-to-back studies share the same novelty and mutually confirm each other's final conclusions, as they employ different species and methods to arrive at similar outcomes. However, our study encompasses a larger scope, as we sequenced more de novo genomes and conducted comprehensive analyses and experimental tests with a systematic comparison of HS biosynthesis and loss in the Solanaceae family. Our findings further suggest the retention of syntenic blocks and key genes within the HS biosynthesis pathway in certain non-HS-producing Solanaceae crops, such as potatoes, tomatoes, tobaccos, and chilies, which are extensively cultivated worldwide. Based on our results, it appears feasible to engineer desired HS biosynthesis into these non-HS-producing Solanaceae crops.

Reviewer #2 (Remarks to the Author):

The authors have done a tremendous job of analyzing the biology of the TRI gene and also performed gain of function experiments in potato.

Secondly, they added measures on wolfberry.

The manuscript has greatly increased in value.

My comment regarding the number of duplications was a cautionary one, in our hand also purging duplicates does not always get rid of all resolved haplotypes leading to duplications. But it is a step in the right direction. A simple solution would be to point out in the discussion that there might be remaining issues.

Reply: Thank you for your appreciation of our work and also excellent suggestion. We have discussed the issues present in this section within the discussion of revised manuscript. Please refer to lines 365-369 in the revised manuscript for details.

Reviewer #3 (Remarks to the Author):

In the revised version of their manuscript describing the evolutionary history of tropane alkaloid biosynthesis in the Solanaceae family, the authors have addressed all my concerns. I have especially appreciated their efforts to generate the gain-of -function mutant and its characterization. The whole story is interesting and the numerous corrections they did considerably improve this version of the manuscript.

Reply: Thank you for your appreciation of our work and the great efforts for reviewing our manuscript.